# Bayesian nonparametric (non-)renewal processes for analyzing neural spike train variability

**David Liu**
Department of Engineering
University of Cambridge
dl543@cam.ac.uk

**Máté Lengyel**
Department of Engineering
University of Cambridge
Department of Cognitive Science
Central European University
m.lengyel@eng.cam.ac.uk

## Abstract

Neural spiking activity is generally variable, non-stationary, and exhibits complex dependencies on covariates, such as sensory input or behavior. These dependencies have been proposed to be signatures of specific computations, and so characterizing them with quantitative rigor is critical for understanding neural computations. Approaches based on point processes provide a principled statistical framework for modeling neural spiking activity. However, currently, they only allow the instantaneous mean, but not the instantaneous variability, of responses to depend on covariates. To resolve this limitation, we propose a scalable Bayesian approach generalizing modulated renewal processes using sparse variational Gaussian processes. We leverage pathwise conditioning for computing nonparametric priors over conditional interspike interval distributions and rely on automatic relevance determination to detect lagging interspike interval dependencies beyond renewal order. After systematically validating our method on synthetic data, we apply it to two foundational datasets of animal navigation: head direction cells in freely moving mice and hippocampal place cells in rats running along a linear track. Our model exhibits competitive or better predictive power compared to state-of-the-art baselines, and outperforms them in terms of capturing interspike interval statistics. These results confirm the importance of modeling *covariate-dependent* spiking variability, and further analyses of our fitted models reveal rich patterns of variability modulation beyond the temporal resolution of flexible count-based approaches.

## 1 Introduction

Analyses of spike time [49, 76, 86] and count [12, 26, 57] statistics have revealed neural responses *in vivo* to be structured but generally variable or noisy [24, 44, 70, 78]. To model this stochastic aspect of neural spike trains, probabilistic approaches based on temporal point processes have been widely applied. This in turn has been a major driver of point process theory development [42] for capturing spiking variability structure with statistical models [74]. The study of neural computation underlying naturalistic behavior in particular involves non-stationary spike trains, which presents a significant challenge as apparent spiking variability is a result of both irreducible "intrinsic" neural stochasticity as well as dependencies on behavioral covariates that can themselves vary on multiple time scales.

Different approaches have been proposed for handling non-stationary spike trains, starting with the classical log Cox Gaussian process [14, 55] to allow variations in the local intensity or firing rate while modeling independent spikes. Dependencies on previous spikes can be captured to first order with renewal processes, and these models have been extended to non-stationary cases through modulation of the hazard function with some time-dependent function [43, 79] or through rescaling interspike intervals with a covariate-dependent rate function [2, 6]. Another approach based on Hawkes processes and spike-history filters [45, 83, 91] introduces conditional point processes that go

37th Conference on Neural Information Processing Systems (NeurIPS 2023).

beyond the first order Markov assumption. Approaches based on recurrent networks [51, 90] and neural ODEs [10, 41] can in theory capture arbitrarily long dependencies on past spikes and input covariates, but provide more limited descriptive interpretability.

However, not only the rate but also the variability of spiking encodes task-relevant information [34, 60], and bears signatures of the underlying computations [11]. Importantly, this variability has stimulus- and state-dependent structure [12, 21, 48, 67]. In statistical modeling language, this corresponds to heteroscedastic or input-dependent observation noise. Such structure reflects computations performed in the underlying neural circuit, and thus characterizing it from data in a flexible and robust manner is critical for advancing theories of neural computation. The classical approaches reviewed above do not attempt to characterize such covariate-dependent changes in variability. Flexible count models have been introduced to more faithfully capture variability at the count level [28], and recent work has extended this to the general case of input-dependent variability [48]. Count approaches however are limited in the resolution of the analysis set by the time bin size. In addition, the resulting count statistics strongly depend on the chosen time bin size [48, 73].

While firing rates are routinely modeled as input-dependent, extending point process models with input-dependent variability has not been widely explored in the literature. Rate-rescaled and modulated renewal processes rely on fixed base renewal densities. Allowing the shape parameters of the renewal density to vary with covariates in the corresponding hazard function is one potential approach, but this still relies on a commitment to a particular parametric family of renewal densities. Spike-history filters in conditional point processes are conventionally fixed and thus do not directly model input-dependent spiking variability, though dependence on observed and unobserved covariates [91] and switching filters based on discrete states [23] have been considered. Recent work has moved away from parametric filters to nonparametric Gaussian processes [18], which can be extended to flexibly model dynamic filters as functions of external covariates with a spatio-temporal Gaussian process. However, any modulation of the filter will no longer permit fast convolutions, and such models will be computationally expensive as the filter needs to be recomputed every time step. The direct nonparametric estimation of conditional intensity functions based on maximum likelihood has been explored [13, 82], but scalable Bayesian approaches have remained absent.

**Contribution**   To enable flexible modeling as well as modulation of instantaneous point process statistics for analyzing neural spike train variability, we introduce the Bayesian nonparametric non-renewal process (NPNR). NPNR builds on sparse variational Gaussian processes and defines a nonparametric prior over conditional interspike interval distributions, generalizing modulated renewal processes with nonparametric renewal densities and spike-history dependencies beyond renewal order. In particular, our point process can flexibly model modulations of not only spiking intensity but also variability. We validate our model using parametric inhomogeneous renewal processes, recovering conditional interspike interval distributions and identifying renewal order in spike-history dependence. On neural data from mouse thalamus and rat hippocampus, our method has competitive predictive power while being superior in capturing interspike interval statistics from the non-stationary data. In particular, our method provides instantaneous measures of spike train variability that are modulated by covariates, and shows rich variability patterns in both datasets consistent with previous studies at coarser timescales. We provide a `JAX` [4] implementation of our method as well as established baseline models within a scalable general variational inference scheme. [1]

## 2   Background

We start with a brief overview of the theoretical foundations and related point process models, as well as their combination with Gaussian processes to introduce non-stationarity.

### 2.1   Temporal point processes

Statistical modeling of events that occur stochastically in time is handled by the general framework of temporal point processes [59, 72]. Denoting the number of events that occurred until time $t$ by $N(t)$, a temporal point process model is completely characterized by its conditional intensity function (CIF)

$$\lambda(t|\mathcal{H}_t) = \lim_{\delta t \to 0} \frac{\mathbb{E}[N(t + \delta t) - N(t)|\mathcal{H}_t]}{\delta t} \tag{1}$$

---

[1]Code available at https://github.com/davindicode/nonparametric-nonrenewal-process

where $\lambda(t)\delta t$ is the probability to emit a spike event in the infinitesimal interval $[t, t + \delta t)$ conditioned on $\mathcal{H}_t$, the spiking history before $t$. We can write the point process likelihood for a single neuron spike train consisting of an ordered sequence of $S$ spike events at times $t_i$ as [5]

$$p(t_1, \ldots, t_S | \lambda(\cdot)) = \left[ \prod_{i=1}^{S} \lambda(t_i | \mathcal{H}_{t_i}) \right] e^{-\int_0^T \lambda(t' | \mathcal{H}_{t'}) \, \mathrm{d}t'} \tag{2}$$

In neuroscience applications, one often wants to describe modulation of the point process statistics with some time-varying covariates $\boldsymbol{x}(t)$, such as animal head direction or body position, which leads to a generalized CIF $\lambda(t | \mathcal{H}_t, \boldsymbol{x}_{\leq t})$. Several classes of models have been proposed that are defined by particular restrictions on the functional form of $\lambda(t | \mathcal{H}_t, \boldsymbol{x}_{\leq t})$.

### 2.1.1 Inhomogeneous renewal processes

**Renewal assumption** The statistical model in Eq. 2 describes dependencies between all spikes. One common simplification is the renewal assumption: interspike intervals (ISIs) $\Delta^{(i)} = t_{i+1} - t_i$ are drawn i.i.d. from an interval distribution called the renewal density $g(\Delta; \theta)$ with parameters $\theta$. This induces a Markov structure $p(t_i | t_{i-1}, t_{i-2}, \ldots) = p(t_i | t_{i-1})$ in the spike train likelihood

$$p(t_1, \ldots, t_S; \theta) = p(t_1) \, p(t_2 | t_1) \cdots p(t_S | t_{S-1}) = p(t_1) \prod_{i=1}^{S-1} g(u_i; \theta) \tag{3}$$

Common renewal densities used for neural data are the exponential (equivalent to a Poisson process), gamma, and inverse Gaussian distributions [2, 6].

**Hazard function modulation** Non-stationary point processes need to model changes in statistics with time, and combined with the renewal assumption one obtains inhomogeneous renewal processes. A classical approach that dates back to Cox [14] is to modulate the hazard function (Appendix A)

$$\lambda(t | t_i) = h(\tau) \cdot \rho(t) \tag{4}$$

with time since last spike $\tau = t - t_i$ and the modulation factor $\rho(t)$. In our context, this can be replaced by some function of covariates $\rho(\boldsymbol{x}_t)$ [43]. A multiplicative interaction between $\rho$ and $h$ as above is typically considered, though this framework allows general parametric forms [71, 79].

**Rate-rescaling** Another approach that has been widely applied in the neuroscience community is rate-rescaling [2, 6], closely related to time-rescaling (see Appendix B.3). Here, modulation is achieved with a rate function $r(\boldsymbol{x}_t) \geq 0$ that transforms time $t$ into rescaled time $\tilde{t}$

$$\tilde{t}(t) = \int^t r(\boldsymbol{x}_{t'}) \, \mathrm{d}t' \tag{5}$$

This maps all spike times $t_i$ to rescaled times $\tilde{t}_i$, and will be one-to-one as long as $r(t) > 0$. By modeling the rescaled ISIs $\tilde{\Delta}^{(i)} = \tilde{t}_{i+1} - \tilde{t}_i$ as drawn from a stationary renewal density $g(\cdot)$, we obtain an inhomogeneous renewal process from a homogeneous one. The CIF becomes dependent on the covariate path since last spike $\mathcal{P}_k = \{\boldsymbol{x}(u) | u \in (t_i, t])\}$, see Appendix B.3.

### 2.1.2 Conditional point processes

**Conditional Poisson processes** The renewal assumption ignores correlations between ISIs, which generally are observed in both the peripheral and central nervous system [1, 25] and can be computationally relevant for signal detection and encoding [8, 9, 22, 69]. Going beyond Markovian dependencies, a tractable approach, similar to Hawkes processes [51], is to introduce a causal linear filter $h(t)$ that is convolved with spikes and added to the log CIF, giving conditional Poisson processes

$$\log \lambda(t | \mathcal{H}_t, \boldsymbol{x}_t) = h * y(t) + f(\boldsymbol{x}_t), \quad \text{with } y(t) = \sum_i \delta(t - t_i) \tag{6}$$

where $*$ denotes temporal convolution. These models are closely linked to mechanistic integrate-and-fire models [52, 85] and have a long history in the neuroscience literature [18, 31, 35, 47, 62, 66, 83], appearing as generalized linear models (GLMs) and spike response models (SRMs).

**Conditional renewal processes** An even more expressive model can be obtained by replacing the Poisson spiking process with a rate-rescaled renewal process. This results in a conditional renewal process, where the rate function has history dependence $r(t|\mathcal{H}_t, \boldsymbol{x}_t)$ as in Eq. 6.

## 2.2 Gaussian process modulated point processes

Gaussian processes (GPs) represent a data-efficient alternative to neural networks, which have been widely used to model the CIF [59, 72, 90]. When combining GPs with point process likelihoods, the resulting generative model leads to doubly stochastic processes for event data. Placing a Gaussian process prior [87] over the log intensity function leads to the classic log Cox Gaussian processes [14, 55], and in the same spirit one can modulate renewal hazard functions [79] or perform rate-rescaling [16] with GPs. Such constructions form the basis of many widely used Bayesian neural encoding models for spike trains, both for modeling single neuron responses [15, 16, 68] as well as population activity [19, 93]. Combining the flexibility offered by renewal and conditional point processes with GP rate or modulation functions within a variational framework has been impeded by the fact that the original papers were built on a GLM framework with parametric covariate mappings [6, 7, 18, 65]. To provide a fair comparison of NPNR to these baselines, we implement a general scalable variational inference framework for the construction and application of such models (see Appendix B for details on the baseline models).

## 3 Method

We now introduce the nonparametric non-renewal (NPNR) process and present the approximate Bayesian inference scheme used for model fitting, noting connections to related works in the literature. Our NPNR model provides a nonparametric generalization of modulated renewal processes beyond renewal order, and adds suitable inductive biases for neural spike train data. It implicitly defines a flexible prior over conditional ISI distributions that can be computed using pathwise conditioning, which enables one to analyze spiking variability modulation with minimal parametric constraints. Furthermore, the Bayesian framework provides an elegant data-driven approach to inferring the lagging ISI order of the spike-history dependence.

### 3.1 Generative model

**Conditional intensity surface priors** To obtain flexible modulated point process models, we directly model the CIF, or more precisely its logarithm, of the form

$$\lambda(t|\mathcal{H}_t, \boldsymbol{x}_{\leq t}) = \lambda(t|t_i, t_{i-1}, \ldots, \boldsymbol{x}_t) \tag{7}$$

where $t_i$ is the most recent spike at current time $t$. First considering the renewal case, we note the spatio-temporal structure in the log CIF using time since last spike $\tau = t - t_i$ is

$$\log \lambda(t|\mathcal{H}_t, \boldsymbol{x}_t) = \log \lambda(t|t_i, \boldsymbol{x}_t) = f(\tau, \boldsymbol{x}_t) \tag{8}$$

which suggests placing a spatio-temporal GP prior on the log CIF to describe a log intensity surface

$$f(\tau, \boldsymbol{x}_t) \sim \mathcal{GP}\left(m(\tau, \boldsymbol{x}), k_t(\tau, \tau') \cdot k_x(\boldsymbol{x}, \boldsymbol{x}')\right) \tag{9}$$

This generalizes the parametric forms of modulation considered in previous approaches [43, 79], in particular allowing modulation of the effective instantaneous renewal density by covariates $\boldsymbol{x}_t$. We can introduce lagging ISIs covariates $\Delta_k(t)$ with lag $k$ as depicted in Fig. 1A to extend the model to a non-renewal process

$$\log \lambda(t|t_i, t_{i-1}, t_{i-2}, \ldots) = f(\tau, \Delta_1(t), \Delta_2(t), \ldots) \tag{10}$$

and for a maximum ISI lag $K$ we denote the lagging ISIs $\boldsymbol{\Delta}_t = [\Delta_1(t), \ldots, \Delta_K(t)]$ to obtain

$$\log \lambda(t|\mathcal{H}_t, \boldsymbol{x}_t) = f(\tau, \boldsymbol{\Delta}_t, \boldsymbol{x}_t) \tag{11}$$

**Inductive biases for neural data** For neural spiking data, there are biological properties to consider for building a more realistic prior. Firstly, neurons have refractory periods immediately following a spike, though in practice neural recordings may not respect this due to contamination in spike sorting [38]. Another potentially useful inductive bias is that changes in the spiking intensity of

neurons fluctuate mostly at shorter ISI timescales [32], whereas at longer delays they tend to be temporally smoother. The latter suggests non-stationary GP kernels to be more suitable for modeling the spike-history dependencies [18]. However, non-stationary kernels do not allow straight-forward use of random Fourier features for evaluating GP posterior function samples at many locations with pathwise conditioning [88, 89]. This in particular is needed to compute the conditional ISI distributions $g(\tau|\ldots)$ in Eq. 15, see Appendix B.5 for details. To achieve the desired non-stationarity for modeling Eq. 11 while maintaining the ability to draw samples using pathwise conditioning, we apply time warping on $\tau$ from $[0,\infty) \to [0,1]$ with some warping timescale $\tau_w$

$$\tilde{\tau} = 1 - e^{-\tau/\tau_w} \quad \Longleftrightarrow \quad \tau = -\tau_w \log(1 - \tilde{\tau}) \tag{12}$$

and place a stationary Gaussian process prior over the warped temporal dimension $f(\tilde{\tau}, \ldots) \sim \mathcal{GP}$ with a temporal kernel $k(\tilde{\tau}, \tilde{\tau}') = k(\tilde{\tau} - \tilde{\tau}')$. This transformation is monotonic (see Fig. 1B), and hence we can easily compute the transformation on the CIF

$$\lambda(t|\ldots) = \lambda(\tilde{t}|\ldots) \left| \frac{\mathrm{d}\tilde{\tau}}{\mathrm{d}\tau} \right| = e^{f(\tilde{\tau}, \ldots)} \frac{e^{-\tau(\tilde{\tau})/\tau_w}}{\tau_w} \tag{13}$$

Similarly, we apply time warping to the $\Delta_k$ dimensions on which we also place stationary kernels $k(\tilde{\Delta}, \tilde{\Delta}') = k(\tilde{\Delta} - \tilde{\Delta}')$. We note that unlike spike-history filters in conditional point processes (Eq. 6) which do not change with inputs, the resulting coupling to past activity in Eq. 11 is dependent on covariates $\boldsymbol{x}_t$. This allows one to capture spiking variability modulation via the conditional ISI distribution perspective discussed below. The refractory nature of real neurons can be addressed by the mean function

$$m(\tilde{\tau}, \boldsymbol{x}) = m(\tilde{\tau}) = a_m \cdot e^{-\tilde{\tau}/\tau_m} + b_m \tag{14}$$

with parameters $a_m, \tau_m, b_m$, where refractory periods can be modeled with large negative $a_m$.

**Conditional ISI distributions**  Instead of looking at the CIF, we can view the model as a prior over conditional ISI distributions as depicted in Fig. 1C using the relation (see Appendix A)

$$g(\tau|\boldsymbol{\Delta}_t, \boldsymbol{x}_{(t_i, t]}) \propto \lambda(t|\mathcal{H}_t, \boldsymbol{x}_t) \cdot e^{-\int_{t_i}^{t} \lambda(t'|\mathcal{H}_{t'}, \boldsymbol{x}_{t'}) \mathrm{d}t'} \tag{15}$$

where one drops the dependence on $\mathcal{H}_t$ in $g$ for the modulated renewal case. If one fixes the lagging ISIs and picks a constant covariate path $g(\tau|\boldsymbol{\Delta}_*, \boldsymbol{x}_*)$, this can be interpreted as an instantaneous ISI distribution of a neuron at the conditioned inputs $\boldsymbol{\Delta}_*$ and $\boldsymbol{x}_*$. Moments of the conditional ISI distribution are computed using Gauss-Legendre quadratures in warped time (Appendix B.5)

$$\mathbb{E}_{g(\tau)}[\tau^m] = \int_0^\infty g(\tau) \, \tau^m \, \mathrm{d}\tau = \int_0^1 \left| \frac{\mathrm{d}\tau}{\mathrm{d}\tilde{\tau}} \right| g(\tau(\tilde{\tau})) \, \tau(\tilde{\tau})^m \, \mathrm{d}\tilde{\tau} \tag{16}$$

and this can be used to compute tuning curves of spike train statistics, such as the mean ISI $\mathbb{E}[\tau]$ and coefficient of variation $\mathrm{CV} = \sqrt{\mathrm{Var}[\tau]}/\mathbb{E}[\tau]$ [58], as a function of $\boldsymbol{x}_*$. This approach generalizes the homogeneous case considered in the literature, and in particular allows one to compute instantaneous measures of non-stationary spike train variability that are otherwise non-trivial to estimate [58, 75].

### 3.2 Inference

**Temporal discretization**  The generative model is formulated as a continuous time model. In practice, neural and behavioural data are typically recorded with finite temporal resolution at small regular intervals $\Delta t$. The cumulative intensity integral has to be approximated by a sum, though note that directly modeling the cumulative hazard function [59] elegantly avoids this for purely temporal point processes. Spike times are now discretized as a binary vector $\boldsymbol{y} = [y_1, \ldots, y_T]$ where $y_t = 1$ if $t$ has a spike event, zero otherwise. Overall, this discretizes the point process likelihood Eq. 2 as

$$p(t_1, \ldots, t_S|\lambda(t)) \quad \to \quad p(\boldsymbol{y}|\boldsymbol{\lambda}) = \prod_{t=1}^{T} \lambda_t^{y_t} e^{-\sum_{t=1}^{T} \lambda_t \Delta t} \tag{17}$$

where we have $T$ time steps in total. Note that the discretization scheme implies we do not have observations at $\tau = 0$, since the time step immediately after an observed spike has $\tau = \Delta t$.

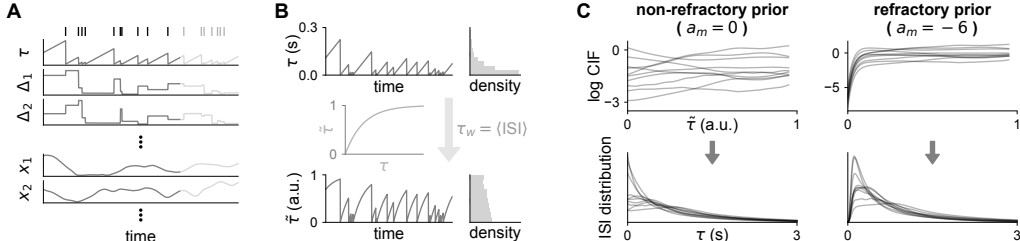

Figure 1: **Schematic of our proposed model. (A)** Time since last spike $\tau$ and lagging ISIs $\boldsymbol{\Delta}$ for an observed spike train (top row) alongside covariates $\boldsymbol{x}$. **(B)** Illustration of the time warping procedure. We fix the warping parameter $\tau_w$ to the empirical mean ISI, which leads to more uniform distributions $\tilde{\tau} \in [0, 1]$ suitable for a stationary GP kernel. **(C)** Prior samples from the generative model for two values of $a_m$ characterized by the lack and presence of a refractory period. The transformation Eq. 15 links the log CIF (top rows) with conditional ISI distributions (bottom rows).

**Variational lower bound** We use stochastic variational inference [39] with batches obtained from consecutive temporal segments and sparse variational GPs [37], giving the loss objective

$$\mathcal{L} = \sum_{n=1}^{N} \left( \mathbb{E}_{q(\boldsymbol{f}_n | \boldsymbol{u}_n)} \left[ -y_{nt} f_{nt} + \Delta t \sum_{t=1}^{T} e^{f_{nt}} \right] + D_{\mathrm{KL}}(q(\boldsymbol{u}_n) | p(\boldsymbol{u}_n)) \right) \tag{18}$$

where $n$ indexes neurons (of which there are $N$)[2], $\boldsymbol{u} = [u_1, \ldots, u_M]$ denotes the set of $M$ inducing points, $p(\boldsymbol{u})$ the GP prior at inducing locations, $q(\boldsymbol{u})$ the variational posterior, and $q(\boldsymbol{f} | \boldsymbol{u})$ the conditional posterior (see Appendix B.1 for details). Combined with temporal mini-batching to fit batch segments of length $T$, we can fit to very long time series given the $\mathcal{O}(N T M^2 + N M^3)$ computational complexity. We also no longer rely on computing hazard functions of parametric renewal densities to obtain the CIF, which can be numerically unstable. Modulated renewal processes instead rely on a specialized thinning procedure [79], but we take a more scalable and general variational approach. Overall, we optimize the kernel hyperparameters, variational posterior mean and covariance, inducing point locations, and mean function parameters $a_m$, $b_m$ and $\tau_m$ using gradient descent with Adam [46] (see Appendix C for details). The time warping parameter $\tau_w$ is fixed in our experiments to the empirical mean ISI, and the hyperparameter $K$ is fixed and chosen in advance (see also subsection on automatic relevance determination below).

**Automatic relevance determination** The Bayesian framework with GPs enables us to perform automatic relevance determination (ARD) over the input dimensions to automatically select relevant input [37, 80]. Applied to lagging ISI dimensions in our NPNR model, this provides an elegant approach to making a data-driven renewal assumption and generally determining the spike-history dependence of the CIF. We choose to fix $\tau_w$ to the empirical mean ISI as shown in Fig. 1B (rather than learning it) to achieve interpretability of kernel timescales for ARD (Fig. 6A) at a small cost of performance (Fig. 12). For a chosen maximum lag $K$, there is no need for manual selection of the history interaction window as for GLM spike-history filters, though recent work on nonparametric GLM filters provides a related window size selection procedure [18]. In the spirit of Bayesian models, we choose $K$ to give a sufficiently high capacity model [40, 80] to be able to flexibly capture history dependence, as seen in panel D of Fig. 3 and Fig. 4.

## 4 Results

All datasets discretize spike trains and input time series at regular intervals of $\Delta t = 1$ ms. We use a product kernel for $k(\boldsymbol{x}, \boldsymbol{x}')$ with periodic kernels for angular dimensions, and squared exponential kernels in other cases. For $k(\tilde{\tau}, \tilde{\tau}')$ and $k(\tilde{\boldsymbol{\Delta}}, \tilde{\boldsymbol{\Delta}}')$, we pick a product kernel with Matérn-$3/2$ (see Fig. 12 for different kernel choices) and set the maximum ISI lag $K = 3$. For illustration, conditional

---

[2]Note the simple summation over $n$ in Eq. 18, as our model does not capture neural correlations without introducing latent covariates [48]. In other words, in its current form, our model treats neurons as independent.

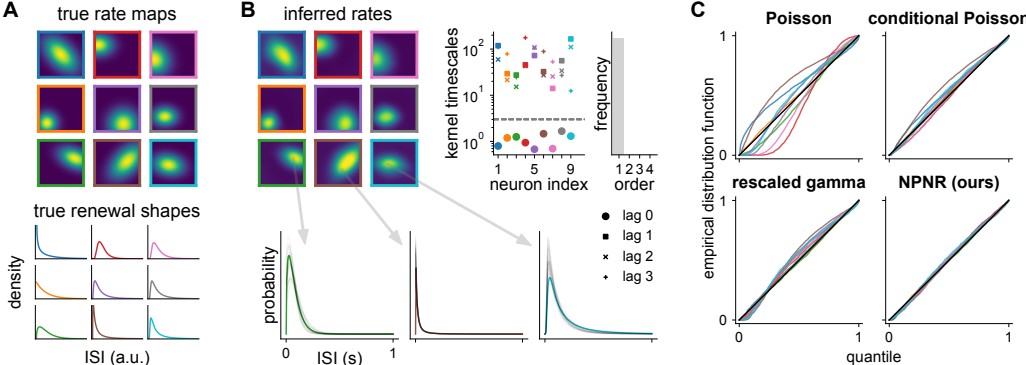

Figure 2: **Validation on synthetic data.** **(A)** True rate maps (brighter is higher) defined over a square environment (top) and base renewal densities (bottom) in each column given by gamma (left), log normal (middle) and inverse Gaussian (right) distributions with various shape parameters. Each color corresponds to a separate neuron. **(B)** Posterior mean rate maps (top left) and conditional ISI distribution samples in gray overlaid on true renewal densities at various locations (bottom) for the NPNR process fit to synthetic data. The relevance boundary (dotted line) for kernel timescales (top right) is placed at $l = 3$ (dimensionless). **(C)** QQ-plots of fitted models (each curve is a neuron).

ISI distributions and corresponding tuning curves are computed by fixing $\mathbf{\Delta}_k$ to be the mean ISI per neuron. Firing rates are defined as $1/\mathbb{E}[\tau]$, since this corresponds to the number of spikes fired per unit time in infinitely large time bins for a renewal process (Eq. 26). GP inducing points were randomly initialized, and for a fair comparison, all models used 8 inducing points for each covariate dimension (including temporal dimensions $\tau$ and $\mathbf{\Delta}$ in the NPNR process). For each experiment, we repeat model fitting with 3 different random seeds and pick the model with the best training likelihood. Further details on experiments are presented in Appendix C.

## 4.1 Validation on synthetic data

For validating our approach, we generate 1000 s of data using rate-rescaling [65] mimicking a place cell population of 9 neurons for an animal moving in a 2D square arena, each with a unique rate map and renewal density (Fig. 2A, details in Appendix C). The models applied are baseline Poisson, raised cosine filter conditional Poisson and rate-rescaled gamma processes (Appendix C), and our NPNR process. Note that the rescaled gamma process is within-model class for 3 of the synthetic neurons. Inferred conditional ISI distributions and rate maps of our NPNR process in Fig. 2B are close to ground truth, showing the ability of our model to capture modulated spiking statistics drawn from various parametric families. To assess how well ISI statistics are captured, we apply time-rescaling using the GP posterior mean functions (Appendix A) which we visualize with quantile-quantile (QQ) plots [6] in Fig. 2C. Again, we see an excellent fit of our model compared to baseline models, indicating that only the NPNR is capable of satisfactorily capturing the empirical ISI statistics. Learned temporal kernel timescales of the NPNR process in Fig. 2B show a clear separation between the time since last spike $\tau$ dimension (lag 0) and lagging ISI $\mathbf{\Delta}$ dimensions (lag $\geq 1$) with the dotted relevance boundary at $l = 3$ (dimensionless), as expected for renewal processes.

## 4.2 Neural data

Now we apply our method to head direction cells in freely moving mice [63, 64] and place cells in rats running along a linear track [54]. We select 33 units from the mouse and 35 units from the rat data, which leads to around 36 and 68 million data points to fit in the training set, respectively (see Appendix C for preprocessing details and Appendix B.2 on data scaling). Experiments involve fitting to the first half of a dataset ($\sim$18 min. for mouse, $\sim$32 min. for rat), and testing on the second half split into 5 consecutive segments. The split into 5 test folds is used to quantify dataset variability. For

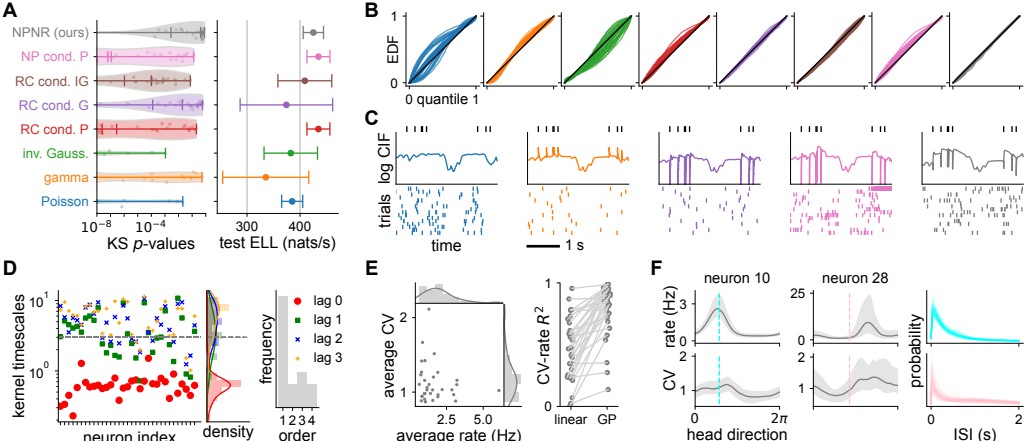

Figure 3: **Application to mouse head direction cell data.** (**A**) Violin plot of KS $p$-values per neuron (left, lines marking quartiles) and test expected log likelihoods with errorbars showing s.e.m. across test folds (right). Larger values in both metrics indicate better model fit to data. (**B**) QQ-plots for various models (each curve is a neuron) identified by color (panel A left). (**C**) Predicted log CIF (middle) for an observed spike train (top) and posterior spike train samples (bottom) conditioned on the same covariates $\boldsymbol{x}_t$ for various models identified by color. (**D**) *Left:* temporal kernel timescales for $\tau$ (lag 0) and $\Delta_k$ dimensions with the relevance boundary at $l = 3$ (dimensionless). *Right:* histogram of "ISI-order" (1 + largest lag $k$ for which $k$ is below the boundary) across neurons. (**E**) Time average of estimated instantaneous rates and CVs from the training data (left) and $R^2$ values of CV-rate regression with a linear and a GP model (right). (**F**) Posterior median and $95\%$ intervals of tuning curves over head direction for the rate and CV, with posterior ISI distribution samples (right) at dashed locations.

prediction, we evaluate the expected log likelihood summed over all neurons

$$\text{ELL} = \sum_n \mathbb{E}_{q(\boldsymbol{f}_n)}[\log p(\boldsymbol{y}_n | \boldsymbol{f}_n)] \tag{19}$$

using Gauss-Hermite quadrature with 50 points (Monte Carlo for renewal processes, see Appendix B). To assess goodness-of-fit to ISI statistics, we compute QQ plots as before and apply the Kolmogorov-Smirnov (KS) test, giving a $p$-value per neuron that indicates how likely the data came from the model (Appendix A). Baselines are the inhomogeneous Poisson (P), rate-rescaled gamma (G) and inverse Gaussian (IG) renewal [6, 83], raised cosine (RC) filter conditional Poisson [66, 85] and renewal [65], and nonparametric (NP) filter conditional Poisson processes [18] (details in Appendix C).

### 4.2.1 Mouse head direction cell data

We choose the animal head direction as our 1D input covariate $x$. From the KS test $p$-value distribution, we see that our model outperforms all baselines in capturing ISI statistics (Fig. 3A left, higher is better; Fig. 3B, QQ plots closer to diagonal). It performs competitively to conditional Poisson processes in terms of predictive performance (Fig. 3A right), but those models fail to capture ISI statistics. In addition, we note the spiking saturation in some samples of the nonparametric conditional Poisson model (Fig. 3C, purple) due to a known instability [3, 29]. Samples from our model (Fig. 3C, gray) exhibit visually similar spike patterns to the real spike train segment. Furthermore, kernel timescales in Fig. 3D show a sizable fraction of the population is characterized by a non-renewal spiking process.

**Neural dispersion regimes**   From Fig. 3E and F, we observe both under- and overdispersion (CV smaller and bigger than one) consistent with a previous study based on spike counts [48]. Estimated instantaneous rates and CVs in Fig. 3E are computed using the conditional ISI distribution evaluated along the time series of covariates in the training data. One can regress instantaneous CV against rate, and the CV-rate $R^2$ shows some cells with near linear relations and some with nonlinear trends that can be captured by a GP. Despite that, many cells still show a low overall $R^2$, implying there is no

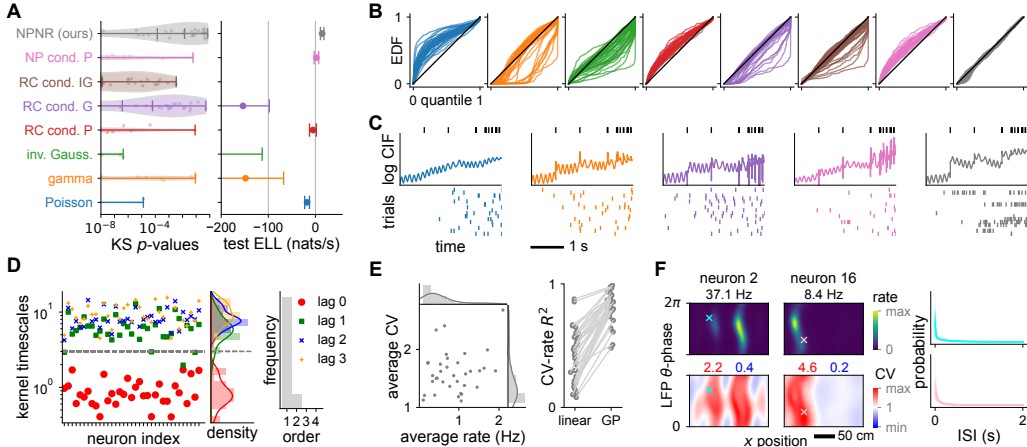

Figure 4: **Application to rat hippocampal place cell data. (A)-(E)** Similar to Fig. 3A-E. **(F)** Posterior mean tuning maps over $x$-position and $\theta$-phase for the rate and CV (left) for left-to-right runs (based on head direction) with posterior ISI distribution samples (right) at marked locations.

parametric relation. Most neurons increase CV with rate, while some show a slight linear decrease reminiscent of refractory Poisson processes (Eq. 31, see Fig. 7 for CV-rate patterns).

### 4.2.2   Rat place cell data

In this case, $x$ is 3D consisting of the body position along the track, head direction, and local field potential (LFP) $\theta$-phase. Our model significantly outperforms all baselines, having both a better KS test $p$-value distribution (Fig. 4A left, higher is better; Fig. 4B, QQ plots closer to diagonal) and predictive performance (Fig. 4A, right). Note that the ELLs for this dataset differ from those shown in Fig. 3A due to the fundamentally different (less predictable) spiking statistics of place cells compared to head direction cells (e.g. due to theta oscillations). As the nonparametric conditional Poisson model introduces nonparametric but *covariate-independent* variability patterns, these results highlights the importance of modeling *covariate-dependent* spiking variability. Note that the rate-rescaled renewal processes struggle to fit this data, with test ELLs of inverse Gaussian models below -200 nats/s (Fig. 4A right). Samples from our model (Fig. 4C, gray) show it captures the characteristic bursting nature of the real spike train segment. Kernel timescales in Fig. 4D show most cells are described well by a renewal process, different to mouse data Fig. 4D.

**Capturing overdispersion**   We see CV values in Fig. 4E higher than the mouse thalamus dataset, consistent with overdispersion of place cell discharge in 2D open field navigation [26]. Similar to the mouse data, the CV-rate $R^2$ again shows there is generally no parametric relation. We also tend to observe larger increases in CV with firing rate compared to mouse data (Fig. 9).

**$\theta$-modulation and phase precession**   Spiking activity modulation during $\theta$-cycles [53] is prominent in rat hippocampus, and is visible here in the log CIF (Fig. 4C). We also see phase precession [76] in Fig. 4F (top), a classical example where spike timing relative to some rhythm has coding significance [33]. Our model enables one to extract not only spiking intensity but also variability, and shows that variability generally inherits the phase precession pattern (Fig. 4F bottom).

## 5   Discussion

### 5.1   Limitations and further work

**ISI statistics**   Apart from the coefficient of variation, there are other ISI statistics that characterize spiking dynamics aspects such as bursting or regularity. Of particular interest is the local coefficient of variation [75], which involves joint statistics of consecutive ISIs $(\Delta^{(i)}, \Delta^{(i-1)})$ that can be computed

from our model (see Appendix D). The same applies to serial correlations [25], which may provide insights into biophysical details [77]. Furthermore, quantifying the shape of ISI distributions is of interest as it is associated with various properties of the underlying neural circuit dynamics [61].

**Neural correlations**    To capture correlations in multivariate spike train data, direct spike couplings as in GLMs are less suitable for current neural recordings compared to latent variable models due to the sparse sampling of populations by electrodes [50]. Combining the latter alongside observed covariates [48] with our point process provides a powerful framework for capturing correlations [81], which can have significant impact on neural coding [56]. To perform goodness-of-fit tests, the Kolmogorov-Smirnov test with time-rescaling can be extended to the multivariate case [30, 92].

## 5.2   Conclusion and impact

We introduced the Bayesian nonparametric non-renewal (NPNR) process for flexible modeling of variability in neural spike train data. On synthetic renewal process data, NPNR successfully captures spiking statistics and their modulation by covariates, and finds renewal order in the spike-history dependence. When applied to mouse head direction cells and rat hippocampal place cells, NPNR has competitive or improved predictive performance to established baseline models, and is superior in terms of capturing ISI statistics, establishing the importance of capturing covariate-dependent variability. NPNR-based analyses recover known behavioral tuning, while also revealing novel patterns of spiking variability at millisecond timescales that are compatible with count-based studies.

Neural firing rates traditionally characterize most computational functions and information encoded by neurons [16, 17, 33], but recent work on V1 [20, 27, 36, 60] and hippocampal place cells [84] have started to assign computationally well-defined roles to variability in the context of representing uncertainty. Our method introduced in this paper is a principled tool for empirically characterizing neural spiking variability and its modulation at the timescales of individual spikes, and we hope our model will be useful for revealing new aspects of neural coding. Such findings are foundational to advances in computational and theoretical neuroscience, and may have downstream practical applications in designing and improving algorithms for brain-machine interfaces.

## Acknowledgments and Disclosure of Funding

This work was supported by the Cambridge European and Wolfson College Scholarship by the Cambridge Trust (D.L.) and by the Wellcome Trust (Investigator Award in Science 212262/Z/18/Z to M.L.). We are grateful to Kristopher Jensen, Marine Schimel and Valentina Njaradi for helpful comments on the manuscript. We would also like to thank Alexander Terenin and Jonathan So for helpful discussions.

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
