# Bayesian nonparametric (non-)renewal processes for analyzing neural spike train variability
# Supplementary Material

## A Point process theory

### A.1 Point process definitions

From the conditional intensity function (CIF) defined in Eq. 1, we can obtain the survival function $S(\tau)$ based on the flux of probability using time since last spike $\tau = t - t_i$

$$\frac{\mathrm{d}}{\mathrm{d}t}\lambda(t) = -\lambda(t)\,S(\tau) \tag{20}$$

which quantifies what the probability is to have no events from $t_i$ up to time $t$. For the relation between the interspike interval (ISI) distribution $g(\tau)$, we require an event to occur within the infinitesimal window $[t, t+\delta t]$ which is proportional to multiplying the CIF with the survival function [14]

$$g(\tau) = \frac{\lambda(t)\,S(\tau)}{\int_{t_i}^{\infty}\lambda(t')\,S(\tau')\mathrm{d}t'} = \frac{\lambda(t)\,e^{-\int_{t_i}^{t}\lambda(t')\,\mathrm{d}t'}}{1 - e^{-\int_{t_i}^{\infty}\lambda(t')\,\mathrm{d}t'}} \tag{21}$$

Another point process quantity that appears in the literature is the hazard function [9, 31], which is related to the CIF by number at risk. In our case of modeling neural spike trains, this is the same as the CIF. The hazard function is often considered in renewal processes, where we have the relation

$$h(t) = \lambda(t|t_i) = \frac{g(\tau)}{1 - \int_0^{t-t_i} g(\tau')\,\mathrm{d}\tau'} = \frac{g(\tau)}{S(\tau)} \tag{22}$$

### A.2 Time-rescaling and Kolmogorov-Smirnov goodness-of-fit test

#### A.2.1 Time-rescaling

Time-rescaling is analogous to Eq. 5 but uses the CIF instead of a rate function

$$\bar{t}(t) = \int^t \lambda(t'|\ldots)\,\mathrm{d}t' \tag{23}$$

with rescaled time $\bar{t}$. The rescaled ISIs should then be exponentially distributed if the ISIs are computed from spike train samples of the point process with the given CIF [27].

#### A.2.2 Quantile-quantile plots and dispersion

The time-rescaled ISIs $\bar{\Delta}$ can be transformed with the cumulative density function of the unit exponential distribution into quantiles in the range $[0, 1]$

$$q(\bar{\Delta}) = F_{\exp}(\bar{\Delta}) = 1 - e^{-\bar{\Delta}} \tag{24}$$

which should be uniformly distributed if we are working with samples from the point process with the given CIF. One can visualize the empirical distribution of quantiles from rescaled ISIs using a quantile-quantile (QQ) plot as in Fig. 2C. There we plot the empirical distribution function $F(q)$ of quantiles $q$. If the point process model matches the empirical ISIs well, the QQ plot curve will follow the diagonal (i.e. the empirical distribution function of a uniform random variables).

The terms over- or underdispersion describe empirical quantile distributions that do not match the point process model. Overdispersion is associated with too many extreme quantile values near $0$ or $1$. On the other hand, underdispersion involves an excess of quantiles close to the median activity $q = 1/2$. These two forms of dispersion will show up characteristically on QQ plots, with underdispersion associated to an S-like curve and overdispersion its reflection along the diagonal. In neuroscience, the reference point process is conventionally taken to be an intensity-matched Poisson process. Over- or underdispersion then refers to to spiking activity that is more irregular or regular temporally than a Poisson process, respectively.

### A.2.3 Kolmogorov-Smirnov test

The intuition from QQ plots can be formalized using a statistical test called the Kolmogorov-Smirnov goodness-of-fit test. The test statistic is defined by

$$T_{\mathrm{KS}} = \max_q |F(q) - q| \tag{25}$$

which represents the biggest vertical distance of the empirical distribution function to the diagonal. Since we deal with a different number of ISIs for each neuron, we report $p$-values of the observed $T_{\mathrm{KS}}$ [30] in Fig. 3A and Fig. 4A to have a comparable measure across neurons. Low $p$-values indicate the null hypothesis (the proposed point process model) should be rejected, or equivalently does not match the empirical data well. In the literature, this is a standard test for assessing goodness-of-fit of neural spike train models [8, 23].

## A.3 Renewal processes

### A.3.1 Firing rates and ISIs

The law of large numbers for renewal processes [24] shows that for a Markov renewal process

$$\lim_{T \to \infty} \frac{N_T}{T} = \frac{1}{\mathbb{E}[\tau]} \tag{26}$$

where $N_T$ is the number of events in the interval $[0, T]$. This relation can be interpreted as the asymptotic firing rate being equal to the reciprocal mean ISI. Note for finite bin sizes $T$, this generally does not hold and the two quantities will differ depending on the ISI distribution shape.

### A.3.2 Parametric renewal density families

Below we give the parametric densities for renewal processes used in the paper. Note that for model fitting, we rescale the density such that we obtain unit mean by dividing $\tau$ with $\mathbb{E}[\tau]$.

**Gamma**    The Gamma renewal process is obtained from picking a Gamma renewal density

$$g(\tau; \alpha) = \frac{1}{\Gamma(\alpha)} \tau^{\alpha-1} e^{-\tau} \tag{27}$$

which is parameterized by the shape $\alpha$. Note the Poisson process corresponds to $\alpha = 1$ i.e. the exponential distribution. This has mean $\mathbb{E}[\tau] = \alpha$. The shape parameter $\alpha$ acts as a "tuning knob" for spiking randomness, with $\alpha < 1$ leading to overdispersed and $\alpha > 1$ underdispersed activity.

**Inverse Gaussian**    The inverse Gaussian is based on the

$$g(\tau) = \sqrt{\frac{\lambda}{2\pi\tau^3}} e^{-\frac{\lambda(\tau-\mu)^2}{2\mu^2\tau}} \tag{28}$$

where the parameter $\lambda$ is absorbed by the rate-rescaling transform. Hence we take $\lambda = 1$ without loss of generality for neuroscience applications. This has mean $\mathbb{E}[\tau] = \mu$.

**Log-normal**    The log-normal distribution is named as the logarithm of the random variable is normally distributed

$$g(\tau) = \frac{1}{\tau\sigma\sqrt{2\pi}} e^{-\frac{(\log\tau-\mu)^2}{2\sigma^2}} \tag{29}$$

noting that the rate-rescaling absorbs the parameter $\mu$. Hence in neuroscience applications, the renewal density will be parameterized by $\sigma$ with $\mu = 0$. This has mean $\mathbb{E}[\tau] = e^{\sigma^2/2}$.

**Refractory Poisson**    The Poisson process is commonly used, but it does not account for the refractory nature of real neurons. A simple modification is to introduce an absolute refractory period

$$g(\tau) = \begin{cases} 0 & \text{for } \tau < \Delta_{\mathrm{abs}} \\ \lambda\, e^{\lambda(\Delta_{\mathrm{abs}}-\tau)} & \text{for } \tau > \Delta_{\mathrm{abs}} \end{cases} \tag{30}$$

which introduces a dependence on the previous spike time, giving a renewal process with

$$\mathbb{E}[\tau] = \Delta_{\mathrm{abs}} + \lambda^{-1}, \qquad \mathrm{Var}[\tau] = \lambda^{-2}, \qquad \mathrm{CV} = 1 - \frac{\Delta_{\mathrm{abs}}}{\mathbb{E}[\tau]} \tag{31}$$

Note the linear relation between the coefficient of variation CV and the inverse mean ISI $\mathbb{E}[\tau]^{-1}$.

### A.3.3 Hazard functions and asymptotic limits

To evaluate the CIF for renewal processes, we need to compute the hazard function as discussed above. Numerically, this can be challenging as we need to compute the fraction of the renewal density and the survival function (Eq. 22) which both tend to 0 for large $\tau$, and hence we compute a truncation of the asymptotic series instead. Note that the survival function for renewal processes $S(\tau) = 1 - C(\tau)$ with $C(\tau)$ cumulative density function of $g(\tau)$.

**Gamma**   The cumulative density function is

$$C(\tau) = \frac{1}{\Gamma(\alpha)} \gamma(\alpha, \tau) \tag{32}$$

where $\gamma(\cdot, \cdot)$ denotes the lower incomplete Gamma function. The survival function satisfies

$$\lim_{\tau \to \infty} S(\tau) = \frac{1}{\Gamma(\alpha)} \tau^{\alpha-1} e^{-\tau} \left( 1 + \sum_{k=1}^{\infty} \frac{\Gamma(\alpha)}{\Gamma(\alpha - k)} \tau^{-k} \right) \tag{33}$$

and the hazard function becomes

$$h(\tau) = \frac{\tau^{\alpha-1} e^{-\tau}}{\Gamma(\alpha, \tau)}, \quad \lim_{\tau \to \infty} h(\tau) = \left( 1 + \sum_{k=1}^{\infty} \frac{\Gamma(\alpha)}{\Gamma(\alpha - k)} \tau^{-k} \right)^{-1} \tag{34}$$

using the upper incomplete $\Gamma(\alpha, x) = 1 - \gamma(\alpha, x)$

$$\lim_{\tau \to \infty} \Gamma(\alpha, x) = x^{\alpha-1} e^{-x} \left( 1 + \sum_{k=1}^{\infty} \frac{\Gamma(\alpha)}{\Gamma(\alpha - k)} x^{-k} \right) \tag{35}$$

**Inverse Gaussian**   The cumulative density function is

$$C(\tau) = \frac{1}{2} \left( 1 + \mathrm{erf}\left( \sqrt{\frac{1}{2\tau}} \left( \frac{\tau}{\mu} - 1 \right) \right) \right) + \frac{1}{2} e^{2/\mu} \left( 1 + \mathrm{erf}\left( -\sqrt{\frac{1}{2\tau}} \left( \frac{\tau}{\mu} + 1 \right) \right) \right) \tag{36}$$

and thus

$$S(\tau) = \frac{1}{2} \left( 1 - \mathrm{erf}(x) \right) - \frac{1}{2} e^{2/\mu} \left( 1 + \mathrm{erf}(x) \right), \quad x = \sqrt{\frac{1}{2\tau}} \left( \frac{\tau}{\mu} - 1 \right) \tag{37}$$

and the survival function has asymptotic limit

$$\lim_{\tau \to \infty} S(\tau) = \frac{1}{2\pi x} e^{-x^2} \left( 1 - e^{2/\mu} \right) \left( \sum_{n=0}^{\infty} \frac{(-1)^n \Gamma(\frac{1}{2} + n)}{x^{2n}} \right) + e^{2/\mu} \tag{38}$$

giving the hazard function limit for $\tau \to \infty$

$$h(\tau) \to \sqrt{\frac{2x^2 \, e^{-2/\mu} \, \tau^{-3}}{e^{-2/\mu} + 2\pi x \, e^{x^2} - 1}} \left( 1 + \frac{e^{-2/\mu} - 1}{e^{-2/\mu} + 2\pi x \, e^{x^2} - 1} \left( \sum_{n=1}^{\infty} \frac{(-1)^n \Gamma(\frac{1}{2} + n)}{x^{2n} \, \Gamma(\frac{1}{2})} \right) \right)^{-1} \tag{39}$$

**Log normal**   The cumulative density function is

$$C(\tau) = \frac{1}{2} \left( 1 + \mathrm{erf}\left( \frac{\log \tau}{\sigma \sqrt{2}} \right) \right) \tag{40}$$

and the survival function has asymptotic limit

$$\lim_{\tau \to \infty} S(\tau) = \frac{e^{-x^2}}{2\pi x} \sum_{n=0}^{\infty} \frac{(-1)^n \Gamma(\frac{1}{2} + n)}{x^{2n}}, \quad x = \frac{\log \tau}{\sigma \sqrt{2}} \tag{41}$$

giving the hazard function

$$h(\tau) = \frac{2}{\tau \sigma \sqrt{2\pi}} \frac{e^{-x^2}}{1 - \mathrm{erf}(x)}, \quad \lim_{\tau \to \infty} h(\tau) = \frac{\log \tau}{\tau \sigma^2} \left( 1 + \sum_{n=1}^{\infty} \frac{(-1)^n \Gamma(\frac{1}{2} + n)}{x^{2n} \, \Gamma(\frac{1}{2})} \right)^{-1} \tag{42}$$

using $\Gamma(1/2) = \sqrt{\pi}$ and the error function expansion

$$1 - \mathrm{erf}(x) = \frac{e^{-x^2}}{\pi x} \sum_{n=0}^{\infty} \frac{(-1)^n \Gamma(\frac{1}{2} + n)}{x^{2n}} \tag{43}$$

# B  Implementation details

## B.1  Sparse variational Gaussian processes

### B.1.1  Gaussian processes as priors over functions

Gaussian processes (GPs) are a class of widely used Bayesian nonparametric models for modeling unknown functions [36]. Briefly, a Gaussian process $\mathcal{GP}(m(\cdot), k(\cdot, \cdot))$ is defined by a mean and covariance function $m(\cdot)$ and $k(\cdot, \cdot)$, and specifies a prior over functions $f(\cdot) \sim \mathcal{GP}$ such that for any set of input locations $\{\boldsymbol{x}_n\}_1^N$, the function values $\boldsymbol{f} = [f(\boldsymbol{x}_1), \ldots, f(\boldsymbol{x}_N)]$ satisfy

$$\boldsymbol{f} \sim \mathcal{N}(\boldsymbol{m}, K) \tag{44}$$

with mean vector $\boldsymbol{m} = [m(\boldsymbol{x}_1), \ldots, m(\boldsymbol{x}_N)]$ and covariance matrix $K_{ij} = k(\boldsymbol{x}_i, \boldsymbol{x}_j)$.

### B.1.2  Sparse approximation

GPs suffer from an $O(N^3)$ computational bottleneck for inference [36], where $N$ is the number of data points. In addition, closed-form inference and prediction are not possible for non-Gaussian likelihoods as used in this paper. To approximate intractable Gaussian process posteriors in non-conjugate settings, as well as remove the $O(N^3)$ bottleneck for inference and posterior evaluation, we use variational inference along with a sparse approximation of the variational psoterior. The latter refers to parameterizing an approximate posterior as a conditional Gaussian distribution, conditioned on $M$ additional function points at locations $\{\boldsymbol{z}_m\}_1^M$ called inducing points. This leads to a sparse posterior in the sense that $M < N$, while the generative model is augmented given by the joint distribution (where we set the mean $m(\cdot) = 0$ for convenience without loss of generality)

$$p(\boldsymbol{f}, \boldsymbol{u}) = \mathcal{N}\left( \begin{bmatrix} \boldsymbol{0}_x \\ \boldsymbol{0}_z \end{bmatrix}, \begin{bmatrix} K_{xx} & K_{xz} \\ K_{zx} & K_{zz} \end{bmatrix} \right) \tag{45}$$

Following [33], we directly parameterize the posterior over the function values at inducing points

$$q(\boldsymbol{u}) = \mathcal{N}(\boldsymbol{\mu}, S) \tag{46}$$

giving a joint posterior for the GP model

$$q(\boldsymbol{f}, \boldsymbol{u}) = p(\boldsymbol{f}|\boldsymbol{u}) \, q(\boldsymbol{u}) \tag{47}$$

The resulting variational free energy or negative evidence lower bound (ELBO) becomes

$$\mathcal{L} = -\mathbb{E}_{q(\boldsymbol{f},\boldsymbol{u})}\left[ \log \frac{p(\boldsymbol{y}|\boldsymbol{f}) \, p(\boldsymbol{f}, \boldsymbol{u})}{q(\boldsymbol{f}, \boldsymbol{u})} \right] = -\mathbb{E}_{q(\boldsymbol{f})}\left[ \log p(\boldsymbol{y}|\boldsymbol{f}) \right] + \mathbb{E}_{q(\boldsymbol{u})}\left[ \log \frac{q(\boldsymbol{u})}{p(\boldsymbol{u})} \right] \tag{48}$$

where the first term is the variational expectation of the negative log likelihood with observations $\boldsymbol{y}$, and the last term is the Kullback-Leibler (KL) divergence of a multivariate normal given by

$$D_{\mathrm{KL}}(q(\boldsymbol{u})\|p(\boldsymbol{u})) = \frac{1}{2}\left( \mathrm{Tr}\big(K_{zz}^{-1}S\big) - M + \boldsymbol{\mu}^T K_{zz}^{-1}\boldsymbol{\mu} + \log \frac{|K_{zz}|}{|S|} \right) \tag{49}$$

which does not involve the inverse of the full covariance matrix $K_{xx}$ as for standard GP inference. Minimizing $\mathcal{L}$ can then be interpreted as finding the inducing points that optimally summarize the training data $\boldsymbol{y}$. The variational expectation term involves the sparse posterior

$$q(\boldsymbol{f}) = \int q(\boldsymbol{f}, \boldsymbol{u}) \, \mathrm{d}\boldsymbol{u} = \int p(\boldsymbol{f}|\boldsymbol{u}) \, q(\boldsymbol{u}) \, \mathrm{d}\boldsymbol{u} \tag{50}$$

which is another multivariate normal with moments

$$\begin{aligned} \boldsymbol{\mu}_{q(\boldsymbol{f})} &= K_{xz} K_{zz}^{-1} \boldsymbol{\mu} \\ \Sigma_{q(\boldsymbol{f})} &= K_{xx} - K_{xz} K_{zz}^{-1} K_{zx} + K_{xz} K_{zz}^{-1} S K_{zz}^{-1} K_{zx} \end{aligned} \tag{51}$$

For predictions at new locations $\{\boldsymbol{x}_*\}$, we obtain the posterior predictive distribution by replacing training locations $\{\boldsymbol{x}_n\}_1^N$ in the sparse posterior above. The decoupling of the data $\boldsymbol{y}$ from the posterior distributions amortizes the variational inference, which allows mini-batching and hence scalability to large data [16]. Overall, this gives the sparse variational Gaussian process (SVGP).

### B.1.3 Posterior sampling and Matheron's rule

Conventionally, one samples from conditionals of multivariate Gaussian distributions by computing the moments and using the Cholesky decomposition. However, this approach comes with $O(N_*^3)$ computational complexity, which becomes prohibitive for evaluating posterior function samples at many locations. An alternative to directly working with distributions is the idea of pathwise sampling or Matheron's rule [37]. For random vectors $\boldsymbol{a}$ and $\boldsymbol{b}$ distributed as a multivariate Gaussian we have

$$(\boldsymbol{a} \mid \boldsymbol{b} = \boldsymbol{\beta}) \stackrel{d}{=} \boldsymbol{a} + \mathrm{Cov}(\boldsymbol{a}, \boldsymbol{b}) \, \mathrm{Cov}(\boldsymbol{b}, \boldsymbol{b})^{-1} \, (\boldsymbol{\beta} - \boldsymbol{b}) \tag{52}$$

where we have now expressed a procedure involving manipulations of samples rather than distributions. The validity follows from the linearity of the transformation and noting the resulting moments match the posterior Gaussian distribution. In particular, $\boldsymbol{a}$ and $\boldsymbol{b}$ are sampled independently.

**Sparse posteriors**  Sparse Gaussian process posteriors are of the form Eq. 50. Matheron's rule applies to sampling from $p(\boldsymbol{f}|\boldsymbol{u})$ with given $\boldsymbol{u}$. In the sparse posterior, we can interpret the expression as drawing $\boldsymbol{\beta} \sim q(\boldsymbol{u})$ and then conditioning $p(\boldsymbol{f}|\boldsymbol{u} = \boldsymbol{\beta})$. This gives us the posterior distribution written as manipulations on a prior sample $f(\cdot) \sim \mathcal{GP}(0, k(\cdot, \cdot))$

$$\boldsymbol{f}_* \mid \boldsymbol{y} \stackrel{d}{=} \boldsymbol{f}_* + K_{*z} \, K_{zz}^{-1} \, (\boldsymbol{\mu} + \boldsymbol{\epsilon}_S - \boldsymbol{f}_z) \tag{53}$$

where $\boldsymbol{f}_*$ and $\boldsymbol{f}_z$ is the same prior sample evaluated at test and inducing point locations, respectively.

**Pathwise conditioning**  From the above, we note that the computational bottleneck is due to sampling the prior $\boldsymbol{f}_*$ at test locations. To work around this, we change our perspective from the function space to the Fourier domain and sample the prior using random Fourier features (see below)

$$\boldsymbol{f}_* \mid \boldsymbol{u} \stackrel{d}{=} \boldsymbol{f}_* + K_{*z} \, K_{zz}^{-1} \boldsymbol{\mu} - K_{*z} \, K_{zz}^{-1} \, (\boldsymbol{f}_z - \boldsymbol{\epsilon}_S) \tag{54}$$

with $\boldsymbol{\epsilon}_S \sim \mathcal{N}(\boldsymbol{0}, S)$. Because Matheron's rule decouples the sampling procedure, we can change the method for sampling the prior to obtain this decoupled posterior sampling procedure. This reduces the computational complexity from $O(N_*^3)$ to $O(N_*)$, combining the relative strengths of the functional and Fourier perspectives of GPs [37].

### B.1.4 Random Fourier features

GPs are conventionally defined in the functional perspective as a prior over functions specified by a covariance kernel function $k(\boldsymbol{x}, \boldsymbol{x}')$. For stationary kernels $k(\boldsymbol{x}, \boldsymbol{x}') = k(\boldsymbol{x} - \boldsymbol{x}')$, and Bochner's theorem states that valid stationary covariance functions must have a non-negative Fourier transform $\tilde{k}(\boldsymbol{\omega}) \geq 0$. One can interpret this as some probability measure $p(\boldsymbol{\omega})$ after suitable normalization, and this provides an alternative view on Gaussian process functions as linear combinations of random Fourier features [26]

$$f(\boldsymbol{x}) = \frac{\sigma}{\sqrt{2L}} \sum_{j=1}^{L} w_j \cos \left( \boldsymbol{\omega}_j \cdot \boldsymbol{x} + \phi_j \right) \tag{55}$$

where $\sigma^2$ is the kernel variance, $w_j \sim \mathcal{N}(0, 1)$, $\boldsymbol{\omega}_j \sim p(\boldsymbol{\omega})$ and $\phi_j \sim \mathcal{U}(0, 2\pi)$. For $L \to \infty$, this will tend to exact function draws from the GP prior. Note for a product kernel as used in this work, $p(\boldsymbol{\omega})$ will also be a product of individual kernel factor probability measures.

**Periodic kernels**  For periodic kernels

$$k(\theta; \sigma, l) = \sigma^2 \, e^{-(1-\cos\theta)/l^2} \tag{56}$$

we use the following series expansion [34]

$$e^{-(1-\cos\theta)/l^2} = \frac{I_0(1/l^2)}{e^{1/l^2}} + \sum_{j=1}^{\infty} \frac{2I_j(1/l^2)}{e^{1/l^2}} \cos\left(j\theta\right) \tag{57}$$

and we note the corresponding probability measure for random Fourier features is discrete. We sample a discrete variable from the truncated series after normalizing the terms

$$p(\omega) = \sum_{j=0}^{J} \frac{c_j}{N_J} \, \delta(\omega - j), \quad N_J = \sum_{j=0}^{J} c_j \tag{58}$$

Due to the discrete nature of the probability measure, sample functions are not directly differentiable as we can no longer rely on the reparameterization trick. We can work around this with importance sampling [5], yielding generalized random Fourier features

$$f(\boldsymbol{x}) = \sigma \sqrt{\frac{N_K}{2L}} \sum_{j=1}^{L} \gamma_j w_j \cos\left(\boldsymbol{\omega}_j \cdot \boldsymbol{x} + \phi_j\right) \tag{59}$$

where now $\boldsymbol{\omega}_j \sim p_{\text{ref}}(\boldsymbol{\omega})$ from a reference distribution where the discrete spectrum factors are not sampled from in a differentiable manner, and with importance sampling weights

$$\gamma_j = \sqrt{\frac{p(\boldsymbol{\omega}_j)}{p_{\text{ref}}(\boldsymbol{\omega}_j)}} \tag{60}$$

By taking $p_{\text{ref}}(\boldsymbol{\omega}) = p(\boldsymbol{\omega})$ we obtain $\gamma_i = 1$ but non-zero gradients for each factor on the backward pass. This is achieved by stopping the automatic differentiation for discrete density factors of $p_{\text{ref}}(\boldsymbol{\omega})$.

### B.1.5 Whitened posteriors

The variable transform $\boldsymbol{v} = L^{-1}\boldsymbol{u}$ leads to a whitened prior $p(\boldsymbol{v}) = \mathcal{N}(\boldsymbol{0}, I)$. Instead of parameterizing $q(\boldsymbol{u}) = \mathcal{N}(\boldsymbol{\mu}, S)$, we can parameterize $q(\boldsymbol{v}) = \mathcal{N}(\boldsymbol{\nu}, W)$ linked by the same variable transform. This variational parameterization generally leads to improved conditioning of the inference problem [1], and we implement SVGPs in all models of this paper with the whitened variational parameterization. The ELBO now involves the KL divergence between the variational posterior and the unit normal distribution. Matheron's rule takes the form with $\boldsymbol{\epsilon}_W \sim \mathcal{N}(\boldsymbol{0}, W)$

$$\boldsymbol{f}_* \mid \boldsymbol{v} \overset{d}{=} \boldsymbol{f}_* + K_{*z} L_{zz}^{-1} \boldsymbol{\nu} - K_{*z} \left(K_{zz}^{-1} \boldsymbol{f}_z - L_{zz}^{-1} \boldsymbol{\epsilon}_W\right) \tag{61}$$

### B.2 General variational inference framework

### B.2.1 Constructing probabilistic models

Constructing the baseline models for this paper with GPs involves different probabilistic model structures specific to each model that are described below. However, the inference framework presented for each model is a special case of general probabilistic programming with stochastic variational inference [4, 12, 19] with SVGPs. From this perspective, the negative ELBO generally consists of two parts

$$\mathcal{L} = \mathcal{L}_{\text{lik}} + \mathcal{L}_{\text{KL}} \tag{62}$$

with $\mathcal{L}_{\text{lik}}$ the variational expectation of the likelihood given the GP posteriors, and $\mathcal{L}_{\text{KL}}$ the KL divergences of each individual GP component.

### B.2.2 Data scaling

Our method builds on sparse variational Gaussian processes, and hence inherits the convergence properties and data scaling from such models analyzed in previous work [7, 16, 25]. In this study, our synthetic validation experiment involves 2D inputs $\boldsymbol{x}_t$ with a million time points, and we are able to accurately recover the ground truth (Fig. 2). Real data has either 1D or 3D $x_t$ and $> 1$ million time points. Based on the synthetic experiment, this suggests that we are in the regime of sufficient data. Note that the total number of GP input dimensions includes the $K = 4$ lagging ISI dimensions for all models.

### B.3 Rate-rescaled renewal processes

### B.3.1 Relation to time-rescaling and modulated renewal processes

Note that rate-rescaling for inhomogeneous renewal processes in Eq. 5 is a special case of time-rescaling when applied to the CIF of the rate-rescaled renewal process. As the CIF of renewal processes is related to the ISI distribution via Eq. 22, we can apply change of variables with Eq. 5 as the transformation. Using the shorthand notation $r(t) = r(\boldsymbol{x}_t)$, we obtain

$$\lambda(t|t_i) = \frac{g(t - t_i)}{1 - \int_{t_i}^{t_{i+1}} g(t' - t_i) \, \mathrm{d}t'} = r(t) \frac{g(\tilde{t} - \tilde{t}_i)}{1 - \int_{\tilde{t}_i}^{\tilde{t}_{i+1}} g(\tilde{t}' - \tilde{t}_i) \, \mathrm{d}\tilde{t}'} = h(\tilde{t}(t)) \cdot r(t) \tag{63}$$

where in the last equality, we wrote in analogous terms to modulated renewal processes Eq. 4. We see that the rate $r$ plays a similar role as the modulator $\rho$, but in addition it also affects the hazard function argument via the rate-rescaling integral $\tilde{t}(t) = \int^t r(\boldsymbol{x}_{t'}) \, \mathrm{d}t'$. This overall gives

$$\lambda(t|\mathcal{H}_t, \mathcal{P}_t) = \lambda(t|t_i, \{\boldsymbol{x}(u)|u \in (t_i, t])\}) \tag{64}$$

which is a covariate path history dependence rather than an instantaneous dependence $\lambda(t|\boldsymbol{x}_t, \ldots)$ like other models in this paper, though in a highly restricted manner through the rate function $r(\boldsymbol{x})$. Note we parameterize $g(\Delta)$ such that the mean is always one (Appendix A.3), which allows $r(t)$ to be interpreted as the instantaneous firing rate of a neuron.

### B.3.2 Time discretization

In discrete time settings, the transformation applied in practice is discrete rate-rescaling [15]

$$\tilde{t} = \sum_{t'=1}^{t} r_{t'} \, \Delta t, \tag{65}$$

Note that time discretization introduces biases in the KS test rate-rescaled renewal process models [13, 15], as the quantiles are computed using the cumulative density function of continuous-time renewal densities. In particular, very small ISIs in the rescaled time domain will be mapped onto the same time bin in the real time domain if the interval is smaller than the bin size. For our nonparametric non-renewal process model, this bias is not an issue due to the flexibility of the CIF function.

### B.3.3 Generative model structure

**Joint versus marginal posterior samples**  The structure in the $\mathcal{L}_{\text{lik}}$ term induced by discrete rate-rescaling Eq. 65 requires us to draw full posterior function samples, as the joint likelihood does not factorize across time steps $p(\boldsymbol{y}|\boldsymbol{f}) \neq p(y_1|f_1) \cdots p(y_T|f_T)$. The posterior rate function samples are then individually integrated over to perform rate-rescaling. Experiments with joint posterior rate function samples from the GP were significantly slower and less numerically stable failing to fit the data successfully, especially on the hippocampal data. Instead, we use a quasi-MAP approximation where we sample from the marginal posterior rates

$$\mathbb{E}_{q(f_1, \ldots, f_T)} \left[ \log p(\boldsymbol{y}|f_1, \ldots, f_T) \right] \to \mathbb{E}_{q(f_1) \cdots q(f_T)} \left[ \log p(\boldsymbol{y}|f_1, \ldots, f_T) \right] \tag{66}$$

Note that this approximation for $\mathcal{L}_{\text{lik}}$ no longer leads to a strict negative ELBO. The marginal posterior samples lead to independent noise in the rates, which gives smaller fluctuations in the rescaled times compared to correlated joint posterior samples.

**Temporal batching and boundary ISIs**  As we fit neural population in parallel and temporally batch with fixed batch sizes, each batch will cut off ISIs at the edges. Generally, the likelihood of a given renewal process has boundary terms to account for unobserved spikes outside the temporal range considered, such as incomplete gamma functions in gamma renewal processes [3] or exponential interval terms to approximate the intractable boundary terms [10, 31]. We instead choose to use the rescaled time from the end of the last batch for initializing the rate-rescaling in a given batch, or ignore the first and last boundary terms of the entire spike train in the first and final batches, respectively. As mini-batching cuts the computation graph between batches, we do not back-propagate beyond the batch cutoff and this introduces a small bias to the gradient as. For sufficiently large batch sizes and overall spike train lengths, these boundary effects are negligible.

**Spike train samples**  To sample from the rate-rescaled renewal processes, one first samples from the homogeneous renewal process to obtain spike times $\tilde{t}_i$. One then applies the transformation shown in Fig. 5A to obtain the real spike times $t_i$, which are modulated according to the rate function $r(t)$.

### B.4 Conditional processes

### B.4.1 Raised cosine spike-history filters

The classical raised cosine basis [35] is defined by

$$h(t; a, c, \phi) = \frac{1}{2} \left[ \cos \left( \min \left( \max \left( a \log \left( t + c \right) - \phi, -\pi \right), \pi \right) \right) + 1 \right] \tag{67}$$

where we can build filters as linear combinations

$$h(t) = \sum_l w_l\, h(t; a_l, c_l, \phi_l) \tag{68}$$

The parameters $a$, $c$ and $\phi$ are fixed while the weights $w$ are learned.

### B.4.2 Nonparametric spike-history filters

Recent work [11] has explored modeling spike-history filters in a nonparametric manner. One can use a Gaussian process with suitable properties for neural data to infer the filter shape in a probabilistic manner. The decaying squared exponential (DSE) kernel encodes suitable inductive biases similar to the ones discussed for our time warping procedure (Eq. 12)

$$k(t, t'; \sigma, \beta, l, l_\beta) = \sigma^2\, e^{-(t-\beta)^2/(2l_\beta^2)}\, e^{-(t'-\beta)^2/(2l_\beta^2)}\, e^{-(t-t')^2/(2l^2)} \tag{69}$$

where $\beta$ and lengthscale $l_\beta$ control the non-stationary aspects of the kernel. Note for $l_\beta \to \infty$, we recover the stationary squared exponential kernel. The GP mean function used is the same as in Eq. 14, now placed on the spike-history time

$$m(t) = a_m\, e^{-t/\tau_m} + b_m \tag{70}$$

### B.4.3 Generative model structure

**Hierarchical GPs** The likelihood term $\mathcal{L}_{\text{lik}}$ now involves performing a 1D convolution on the spike train. For the nonparametric filters, we sample a filter function $h(t)$ from the GP posterior and use this same sample throughout the spike train convolution. To handle the resulting intractable hierarchical model, SVGPs are used within the general variational inference framework discussed in Appendix B.2. Note this involves adding an additional KL divergence term of the spike-history filter GP to $\mathcal{L}_{\text{KL}}$. The parametric raised cosine basis filters simply use point estimates of their parameters, which are treated as hyperparameters of the likelihood.

**Spike train samples and temporal batching** Temporal batching is straight-forward here as the point process likelihood factorizes across time given the CIF, which we can compute if we condition on the past spike train window. Related to this, we can generate spike trains from this model by using an initial spike-history segment and sampling autoregressively using the CIF.

## B.5 Bayesian nonparametric non-renewal processes

### B.5.1 Time warping

For time warping in Eq. 12, we set $\tau_w$ as the empirical mean ISI from the training dataset. The time-warped mean function Eq. 14 becomes in the warped domain

$$m(\tilde{\tau}) = a_m\, (1 - \tilde{\tau})^{\tau_w/\tau_m} + b_m \tag{71}$$

### B.5.2 Computing conditional ISI distributions and tuning curves

To compute the conditional ISI distribution Eq. 15, we need to compute the integrals appearing in Eq. 21, now conditioned on inputs denoted by $\lambda(t|\ldots)$, to obtain a normalized conditional ISI distribution. We define the integral $\Lambda(t, t_i|\ldots)$ which we transform into the warped time domain

$$\Lambda(t, t_i|\ldots) = \int_{t_i}^t \lambda(t'|\ldots)\, \mathrm{d}t' = \int_{\tilde{t}_i}^{\tilde{t}} \lambda(\tilde{t}'|\ldots)\, \mathrm{d}\tilde{t}' = \int_0^{\tilde{\tau}(t-t_i)} e^{f(\tilde{\tau}',\ldots)}\, \mathrm{d}\tilde{\tau}' \tag{72}$$

where for $t \to \infty$, $\tilde{\tau}(t - t_i) \to 1$. This then allows us to write the normalized Eq. 15

$$g(\tau|\ldots) = \frac{\lambda(t|\ldots)\, e^{-\Lambda(t, t_i|\ldots)}}{1 - e^{-\Lambda(\infty, t_i|\ldots)}} \tag{73}$$

The integrals are evaluated using rescaled Gauss-Legendre quadrature points, as the integral limits cannot exceed $[0, 1]$ using the time warping transform. Standard Gauss-Legendre quadratues is defined for the interval $[0, 1]$, and for smaller intervals we linearly scale the quadrature locations

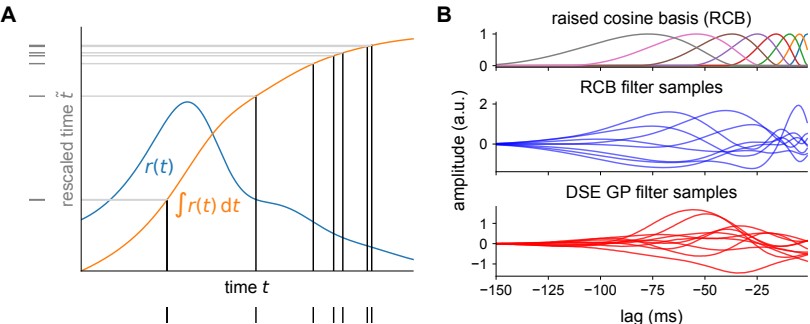

Figure 5: **Details on baseline models. (A)** Rate-rescaling procedure. The cumulative integral (orange) of a non-negative rate function (blue) define the monotonic mapping between real time $t$ and warped time $\tilde{t}$. **(B)** Spike-history filters in conditional point processes. The raised cosine basis (top) is used to construct filters describing interactions with past spikes (middle). Nonparametric GP filters using the DSE kernel encode a similar temporal structure without explicit parameterization (bottom).

while multiplying the weights by the inverse scaling factor. Note the normalization factor in the denominator is independent of $\tau$ and simply uses the standard Gauss-Legendre quadrature method.

For a given set of target values $\tau_*$ where we want to evaluate $g(\tau)$, we need to compute the integrals at a different set of rescaled quadrature points for each target value. This requires us to evaluate a single GP posterior sample $f(\tilde{\tau}, \ldots)$ at many locations, which we achieve using pathwise conditioning (see Appendix B.1 for details). In particular, to compute samples of $g(\tau)$ we need to use the same posterior sample for both the numerator and denominator in Eq. 73.

We generally have many $\tau_*$ values that may only be slightly different, which leads to a large number of rescaled quadrature point locations that are very close to each other. To make this computationally more efficient, we compute the GP posterior samples $f(\tilde{\tau}, \ldots)$ on a regular grid $[0, 1]$ with $G$ grid points and store this as a buffer of pre-computed posterior sample values. We then use linear interpolation to obtain $f(\tilde{\tau}, \ldots)$ at all the required quadrature locations. Overall, this reduces the required GP posterior function sample evaluations to the $G$ grid locations as well as the standard quadrature locations for the normalization factor in Eq. 73. Using a fine enough grid will lead to more accurate values from the interpolation at the cost of a computing and storing a larger buffer.

The moments of $g(\tau)$ can be computed by using an outer loop of rescaled Gauss-Legendre quadratures similar to above with the integrand as given in Eq. 16. Tuning curves of the moments as shown in Fig. 3F and Fig. 4F were computed from joint tuning curve samples, which require evaluating the same posterior ISI distribution sample conditioned on different covariate values. Obtaining all required GP posterior function sample evaluations in a single call would lead to out-of-memory issues, and instead we achieved this using multiple GP posterior function sample evaluation calls while keeping the same pseudo-random number generator key.

### B.5.3 Generative model structure

Sampling from the generative model. Since the history dependence is now contained in the input covariates $\mathbf{\Delta}$, inference is the same as for inhomogeneous Poisson point processes when conditioned on the full input, and the corresponding loss Eq. 18 is naturally amenable to temporal mini-batching. Note that time warping is a feedforward transformation on the input and hence does not affect temporal batching.

### B.6 Code

The code is written in Python and utilizes `JAX` [6] for performing automatic differentiation and numerical optimization for model fitting. All models were implemented from scratch, and we utilize `equinox` [17] for maintaining readability and elegance of the code. In our code base repository, we provide the code and scripts for reproducing all results in this paper.

## C  Further details on experiments

### C.1  General information

**Vectorization across neurons**   Even though our model is designed for single neuron spike trains, we fit neural populations in parallel by vectorizing over all neurons. Each neuron has its own copy of model (hyper)parameters. This implementation also allows straight-forward incorporation of shared latent variables for modeling neural correlations.

**Optimization hyperparameters**   We use the Adam [18] optimizer, where the learning rate is set to $10^{-2}$ at the start and annealed down to $10^{-4}$ with a decay factor per epoch of 0.998 for synthetic experiments and 0.98 for real data experiments. The stopping criterion for model fitting is when 100 epochs have elapsed and the average training loss in each epoch has decreased less than $10^0$ for synthetic and rat data experiments, and $10^1$ for mouse data experiments. We also set a maximum number of 3000 epochs, though in practice most experiments finish before this maximum is reached. We use a temporal batch size of 10000 time points (10 s) for all models except rate-rescaled models where we use a batch size of 30000 time points (30 s). Experiments are run with single floating point precision, and a jitter of $10^{-6}$ is used to stabilize the Cholesky decomposition in GPs.

**Parameter constraints**   Kernel lengthscales, variances and other positive parameters are parameterized as unconstrained parameters pushed through softplus or exponential transformations. Variational distribution covariances are parameterized with a lower triangular matrix where we enforce positive diagonal elements at least as big as the jitter value used. Shape parameters for renewal densities were additionally constrained to be $\alpha \geq 10^{-1}$ for gamma and $\mu \geq 10^{-5}$ for inverse Gaussian renewal densities to ensure numerical stability.

**Variational expectation evaluation**   During training, we use 20 Gauss-Hermite quadrature points to evaluate the variational expectation for all models except rate-rescaled renewal processes. There, we use one Monte Carlo sample from the marginal posterior of the GP rate function to estimate the variational expectation of the likelihood (see Appendix B.3 for a discussion on the GP marginal versus joint posterior sampling for rate-rescaling). For evaluating models on test data, we use 50 Gauss-Hermite quadrature points to evaluate the variational expectation terms and 10 Monte Carlo samples for the rate-rescaled renewal processes.

**Inducing point initialization**   Randomly initialized inducing points by picking random locations without replacement from a $D$-dimensional regular grid on the input covariate space. Inducing point locations are learned as part of the hyperparameters. Note that sparse GPs can suffer from numerical instabilities when some inducing points get very close to each other and cause problematic conditioning numbers, which tend to arise when the total number of inducing points is high [32].

**Conditional process hyperparameters**   For the raised cosine filters in conditional processes (Appendix B.4), we follow [35] and use a 150 ms window with the parameters $a = 4.5$, $c = 9$, and $\phi = \{10, 10 + \frac{1}{7}, 10 + \frac{2}{7}, \ldots, 20\}$ for defining our raised cosine basis consisting of 8 basis functions (Fig. 5B top). Note these parameters are fixed and not optimized during training. The nonparametric conditional Poisson process models spike-history filters using the GP defined in Appendix B.4. Hyperparameters of the kernel are learned during model fitting. We use 6 inducing points initialized uniformly along the 150 ms time lag axis of the spike-history filter.

**Computing resources and reproducibility**   Experiments were ran for training each model or performing analysis of fitted models on single NVIDIA GeForce RTX 2080 Ti GPUs with 11 GB of memory. Each model fitting run can take up to 6 hours on this hardware, but is generally a lot shorter due to the termination criterion. Analysis scripts of fitted models were also run on the same hardware and can take up to 8 hours for the most compute-intensive tasks. The code provided contains a bash script with the exact commands and random seeds that were run to generate the results in this paper. Datasets were taken from the public online database `https://crcns.org/`, with specific instructions on which datasets to use and preprocessing scripts provided in the code repository.

**Neural data preprocessing**   Electrophysiological neural recordings typically provide spike times at higher temporal resolution than animal behavior, the latter being inferred from video recordings

of the animal. To match the input covariates and spike trains in our discrete time point process, we upsample the behavior using Akima interpolation, which suffers less from oscillations when sharp changes are present in the time series [2]. This leads to data sampled at 1 ms regular intervals. From the spike trains, we also compute lagging ISIs at this temporal resolution. Note they include a section at the start of the neural recording where we have undefined lagging ISIs of higher order. To obtain the dataset segment we use, we cut out the section at the start until we have defined lagging ISIs up and until order $K = 4$. We also correct for duplicate spike counts in the 1 ms bins, which are artifacts of spike sorting errors. The number of such duplicate spike counts was minimal, as we also selected for cells which had a fraction of 2 ms refractory period violations less than $5\%$.

### C.2 Validation on synthetic data

For the synthetic population shown in Fig. 2A, the densities used were (each column, top to bottom) the gamma with $\alpha = \{0.5, 1.0, 1, 5\}$, the log normal with $\mu = \{0.5, 1.0, 1, 5\}$, and the inverse Gaussian with $\sigma = \{0.5, 1.0, 1, 5\}$ (equations given in Appendix A.3). Though the Poisson case (left middle dark green) stricly has no past spike dependence in the CIF (i.e. independent of $\tau$), the time warping shape requires nonlinear tuning of the CIF to $\tilde{\tau}$.

We use 16 GP inducing points for baseline models and 48 for our NPNR process. Firing rates and CVs from the NPNR model were computed using 30 posterior samples. For computing the condition ISI distributions, we used $G = 1000$ grid points and 100 Gauss-Legendre quadrature points (see Appendix B.5).

### C.3 Neural data

Violin plots of KS $p$-values in Fig. 3A and Fig. 4A show kernel density estimates using the Silverman method [29]. Instantaneous firing rates and CVs as well as tuning curves of the NPNR model are computed using 50 posterior samples. For computing the condition ISI distributions, we use $G = 1000$ grid points and 100 Gauss-Legendre quadrature points (see Appendix B.5). For calculating the coefficient of determination shown in Fig. 3E and Fig. 4E, we use linear regression on the instantaneous rate and CV estimates (posterior means)... Standard SVGP regression [16] using 8 inducing points with a squared exponential kernel is applied to the log rate and CV, where the logarithmic transform helps to account for the non-stationary lengthscale of CV-rate scatter cloud shapes along the rate axis.

**Neuron selection criteria**  For selecting putative neurons in the neural recordings, we use the mutual information (MI) per spike, tuning curve coherence and tuning curve sparsity measures [38], which are computed from simple histograms of spiking activity and occupancy over binned relevant covariates. For each bin $i$, we compute the number of time steps $T_i$ when the covariates $x_t$ are located in that bin. We also compute the total number of spikes $N_i$. Now we can compute the average rate $r_i$ and relative occupancy distribution $P_i$

$$r_i = \frac{N_i}{\Delta t\, T_i}, \qquad P_i = \frac{T_i}{\sum_i T_i} \tag{74}$$

with time bin size $\Delta t$. From the rate map $r_i$, we obtain a smoothed rate map $r_i^s$ which is given by the convolution of a smoothing kernel with $r_i$. Using these quantities, we get the tuning measures

$$\text{MI} = \sum_i P_i\, r_i \log \frac{r_i}{\sum_j P_j\, r_j}, \qquad \text{coherence} = \frac{\langle r_i r_i^s \rangle}{\sigma_r \sigma_{r^s}}, \qquad \text{sparsity} = 1 - \frac{(\sum_i P_i\, N_i)^2}{\sum_i P_i\, N_i^2} \tag{75}$$

where $\langle \ldots \rangle$ denotes the average and $\sigma_{...}$ is the standard deviation. Note the coherence is identical to the Pearson correlation coefficient between the smoothed and original histogram rate maps.

**Mouse head direction cell data**  We pick the wake portion in session "140313" of mouse "28" from the mouse dataset [21, 22], which involves a freely moving mouse foraging for food with neural recordings from anterodorsal nucleus post-subiculum. Putative head direction cells are selected based on refractory violation fraction $< 2\%$, head direction mutual information $> 0.5$ bits per spike and head direction tuning curve sparsity $> 0.2$. These tuning measures are computed using a binning of head direction into 60 equal bins. Overall, we end up with 1085970 time points for training data. As

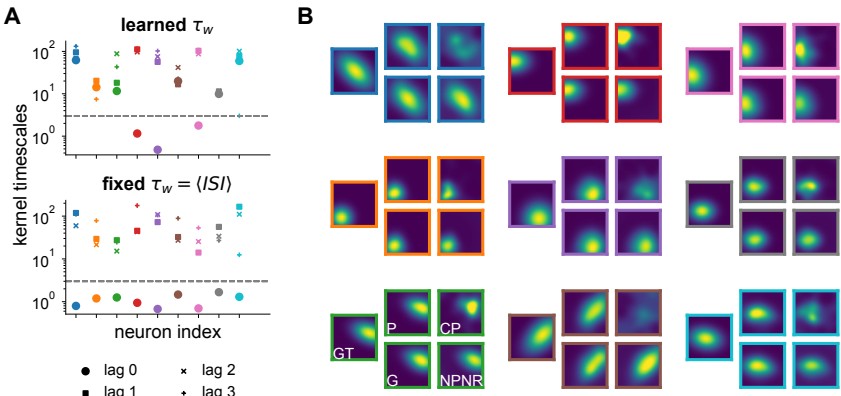

Figure 6: **Additional results from synthetic experiments. (A)** Comparing history-dependence ARD for fixed and learned time warping parameter $\tau_w$. The interference between $\tau_w$ and the kernel timescales in our model causes failure of ARD when learning $\tau_w$, as kernel timescales are not longer identifiable with effective temporal fluctuations in the log CIF. **(B)** Estimated rat maps from fitted models. Each subpanel shows rate maps (brighter is higher) for one synthetic neuron, with the ground truth (GT) on the left and grouped on the right inferred maps from the Poisson (P), conditional Poisson with raised cosine filters (CP), rate-rescaled Gamma (G) and the NPNR model.

we have one dimension (head direction) for our behavioural covariates, we use 8 GP inducing points for baseline model GPs and 40 for our NPNR process (additional $K = 3$ lagging ISI dimensions plus the $\tau$ dimension).

**Rat place cell data**    We pick session "ec014.468" from the rat dataset, which involves the animal running along a linear track of 250 cm with neural recordings from hippocampal area CA1 using silicon-based electrodes [20]. Putative place cells are selected based on refractory violation fraction $< 2\%$, joint $x$-$\theta$ mutual information $> 0.5$ bits per spike and joint $x$-$\theta$ tuning curve sparsity $> 0.6$, coherence $> 0.4$, minimum number of spikes $> 900$ and more than 5 spikes in the first 100 s of the dataset segment used. These tuning measures are computed using a binning of $x$ and $\theta$ into 40 and 30 equal bins, respectively. The smoothing kernel for computing the coherence is a 2D factorized Gaussian with standard deviations of 2 bins in each dimension. Overall, we end up with 1934126 time points for training data. We use 32 inducing points for baseline model GPs and 64 for our NPNR process (additional $K = 3$ lagging ISI dimensions plus the $\tau$ dimension). Note that this uses slightly more than 8 inducing points per dimension of $\boldsymbol{x}$, which we choose to help the model capture potentially more complicated tuning curves relevant to $\theta$-phase precession patterns. The GP jitter is increased to $10^{-5}$ for the nonparametric conditional Poisson process as it is less stable numerically on this dataset.

# D    Additional results

## D.1    Validation on synthetic data

**Test expected log likelihoods**    On synthetic data, the expected log likelihood (ELL) values on test data consisting of 5 different held-out synthetic datasets generated from the same ground truth model are shown in Table 1. This shows that our model outperforms the (conditional) Poisson baselines, while performing comparable to the Gamma renewal process (which is within model class for three of the synthetic neurons) within dataset variability. Overall, these numbers are consistent with the goodness-of-fit results shown in Fig. 2C.

**Automatic relevance determination**    The time warping component of our model has a parameter $\tau_w$, which sets the time range of $\tau$ that is mapped into the range of $\tilde{\tau}$ where the stationary GP can meaningfully model temporal fluctuations. In the main text, we propose to look at kernel timescales

Table 1: Test expected log likelihoods of models on synthetic data.

| Model | Test ELL (nats / s) |
|---|---|
| Poisson | $40.55 \pm 2.09$ |
| Gamma renewal | $47.07 \pm 2.10$ |
| conditional Poisson | $40.82 \pm 2.22$ |
| NPNR (ours) | $44.93 \pm 2.30$ |

to determine relevance of lagging ISI dimensions. However, $\tau_w$ also interferes with the effect of the kernel timescales on the effective timescale in the $\tau$ and $\boldsymbol{\Delta}$ dimensions of the log CIF. Indeed, we see that learning $\tau_w$ generally leads to loss of interpretability for the GP kernel timescales in Fig. 6A. By fixing it to the empirical mean ISI for each neuron, we obtain a suitably normalized time warping scheme where the GP kernel lengthscales are identifiable with temporal fluctuations of the log CIF, and hence the relevance of lagging ISI dimensions.

**Rate map estimation** Visually, the inferred rat maps for the Poisson, rate-rescaled Gamma and nonparametric non-renewal (NPNR) models are quite close to the ground truth. The conditional Poisson model has estimated rate maps that do not match the truth, showing the complicated effects of the spike-history filter on the effective firing rate and smoothness of the rate profile.

## D.2 Neural data

**Head direction cell data** We plot tuning curves in Fig. 8 for the rate and CV for all neurons in our processed dataset. Each time point of posterior mean estimated instantaneous rates and CVs is plotted as a dot in Fig. 7, where we only show the estimates statistics per 10 ms interval.

**Place cell data** We perform the same analysis as for head direction cells, but now the tuning curves are 2D maps over $x$-position and $\theta$-phase. In addition, the animal can run left-to-right or vice versa, which we model by including the head direction covariate of the animal. Tuning curves in Fig. 10 and Fig. 11 are then computed conditioned on the head direction for each run direction. Each time point of posterior mean estimated instantaneous rates and CVs is plotted as a dot in Fig. 9, where we only show the estimates statistics per 10 ms interval.

**Temporal kernel selection** We compare Matérn kernels of different order to use as temporal kernels in our model, as well as fixing the time warping parameter $\tau_w$. From Fig. 12, we see that the differences in performance are small. Learning $\tau_w$ generally bumps up the predictive performance, but has mixed effects on the KS $p$-value distributions. The choice made in the main paper (Matérn-$3/2$ for both $k(\tilde{\tau})$ and $k(\tilde{\boldsymbol{\Delta}})$) gives good performance in both predictive power and KS $p$-values across both neural datasets.

## D.3 Additional ISI statistics and variability measures

Our method captures joint lagging ISI statistics from data into $g(\tau|\boldsymbol{\Delta}, \ldots)$. This provides a more general approach to computing correlations between consecutive rate-rescaled ISIs [23]. As a specific example, the local coefficient of variation [28] can be computed form our model as

$$\mathrm{LV} = 3 \left\langle \left( \frac{\Delta^{(i-1)} - \Delta^{(i)}}{\Delta^{(i-1)} + \Delta^{(i)}} \right)^2 \right\rangle \quad \rightarrow \quad 3 \int_0^\infty \mathbb{E}_{g(\tau|\Delta_1)} \left[ \left( \frac{\Delta_1 - \tau}{\Delta_1 + \tau} \right)^2 \right] \mathrm{d}\Delta_1 \qquad (76)$$

which we can perform using Gauss-Legendre quadratures and time warping similar to Eq. 16, but using a 2D quadrature grid as we now deal with a double integral.

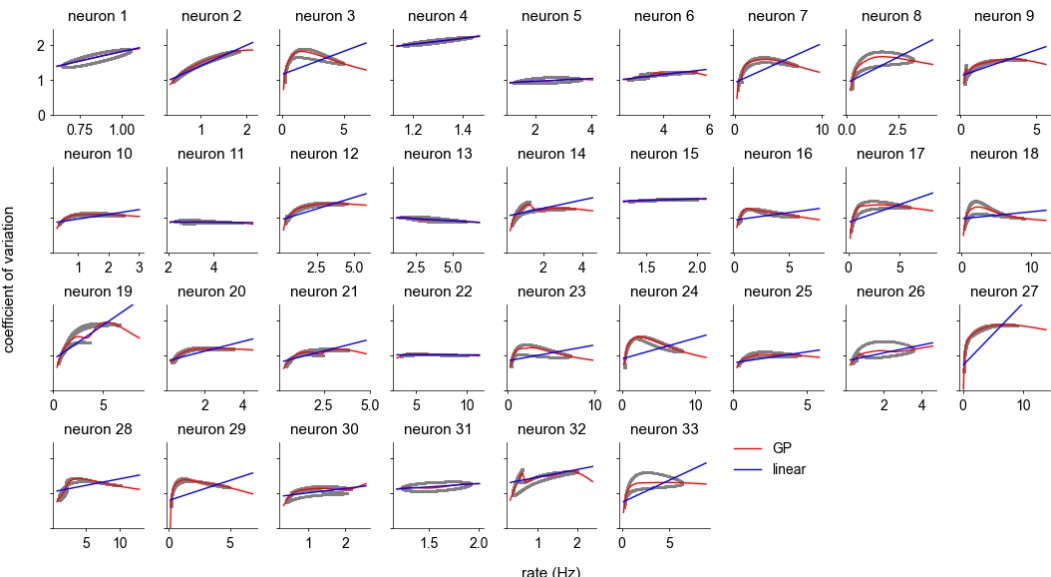

Figure 7: **Spiking variability of head direction cells.** Dots represent estimated instantaneous statistics computed using the posterior conditional ISI distributions. Linear and GP regression fits are shown overlaid on top of the dots. Note the one-dimensional trajectory that is traced out is due to the fact that we have 1D covariates $x_t$ (head direction).

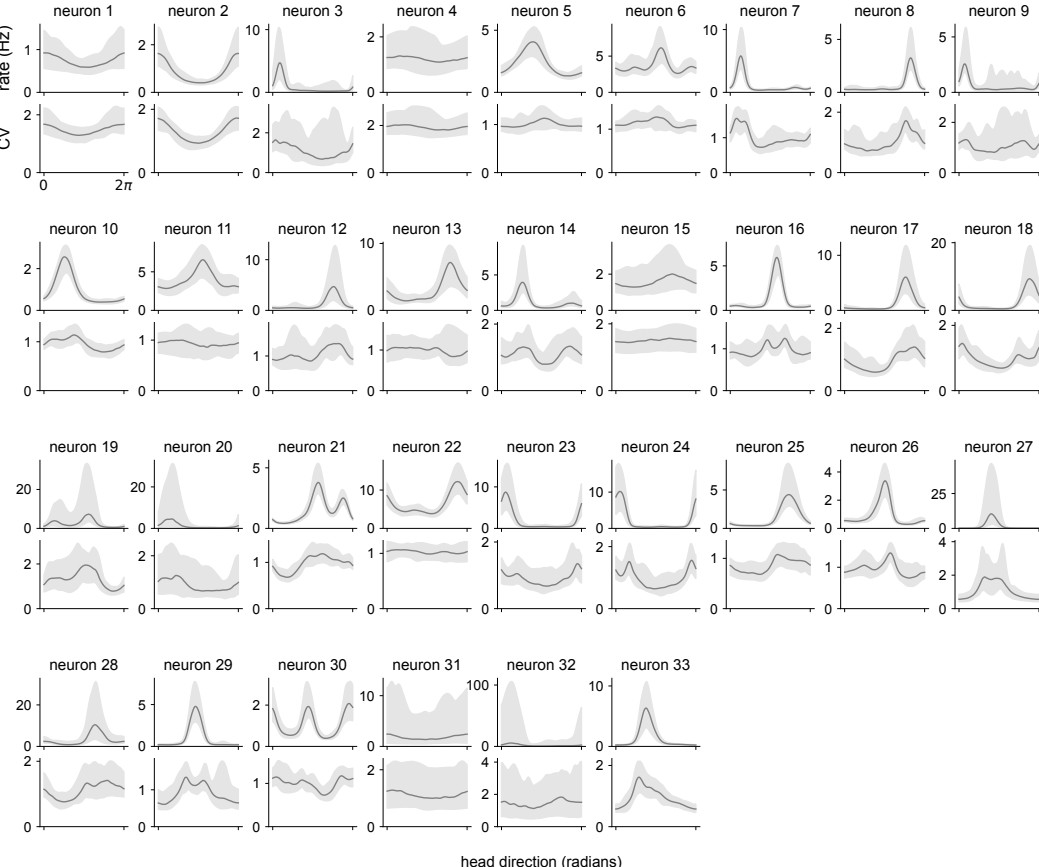

Figure 8: **Tuning curves of head direction cells.** Spike train statistics are computed from conditional ISI distribution samples. Lines show posterior medians, and shaded areas show $95\%$ credible intervals.

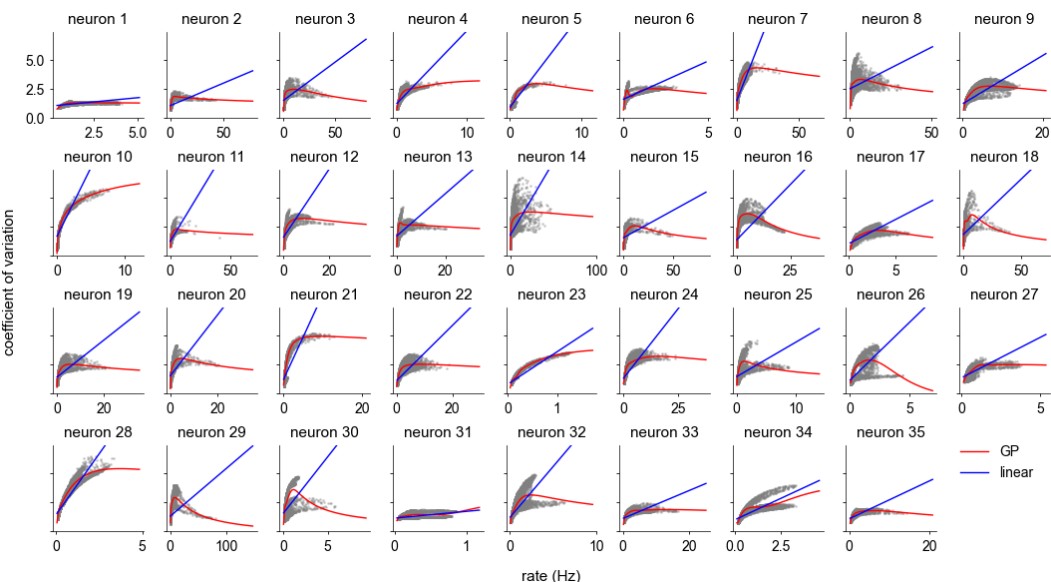

Figure 9: **Spiking variability of place cells.** Dots represent estimated instantaneous statistics computed using the posterior conditional ISI distributions. Linear and GP regression fits are shown overlaid on top of the dots.

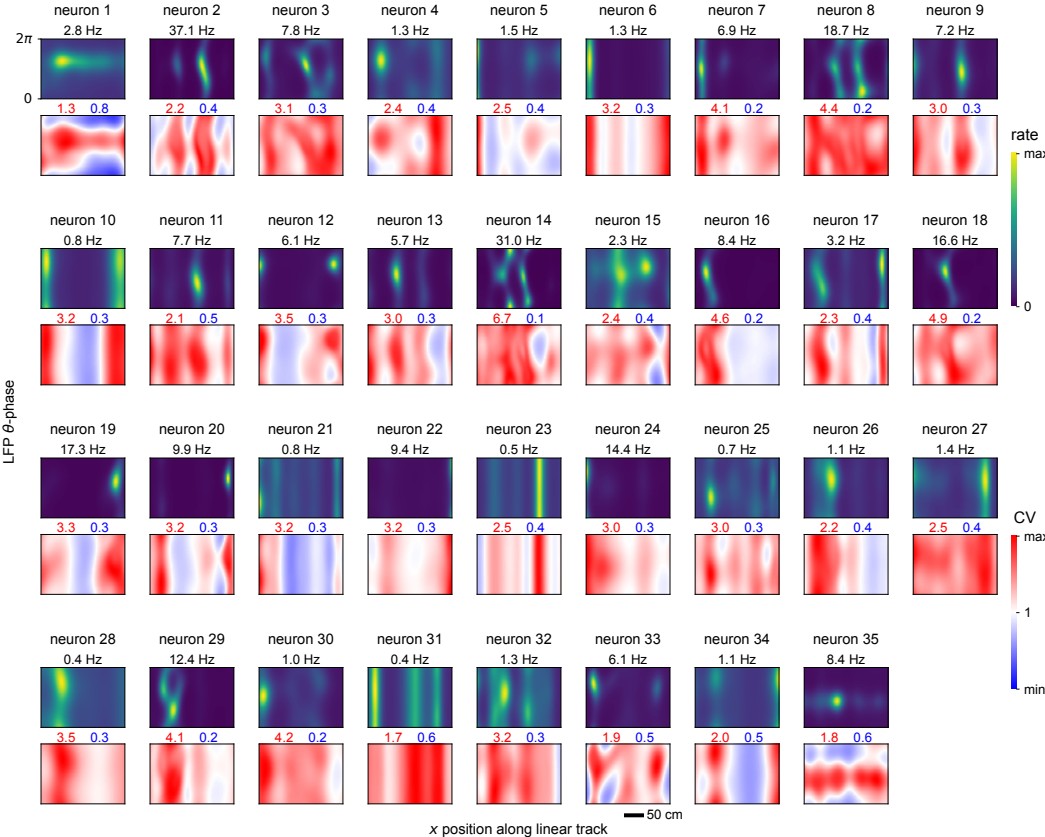

Figure 10: **Tuning curves of place cells for rat running left-to-right.** Heat maps show posterior mean values of spike train statistics computed from the conditional ISI distribution samples.

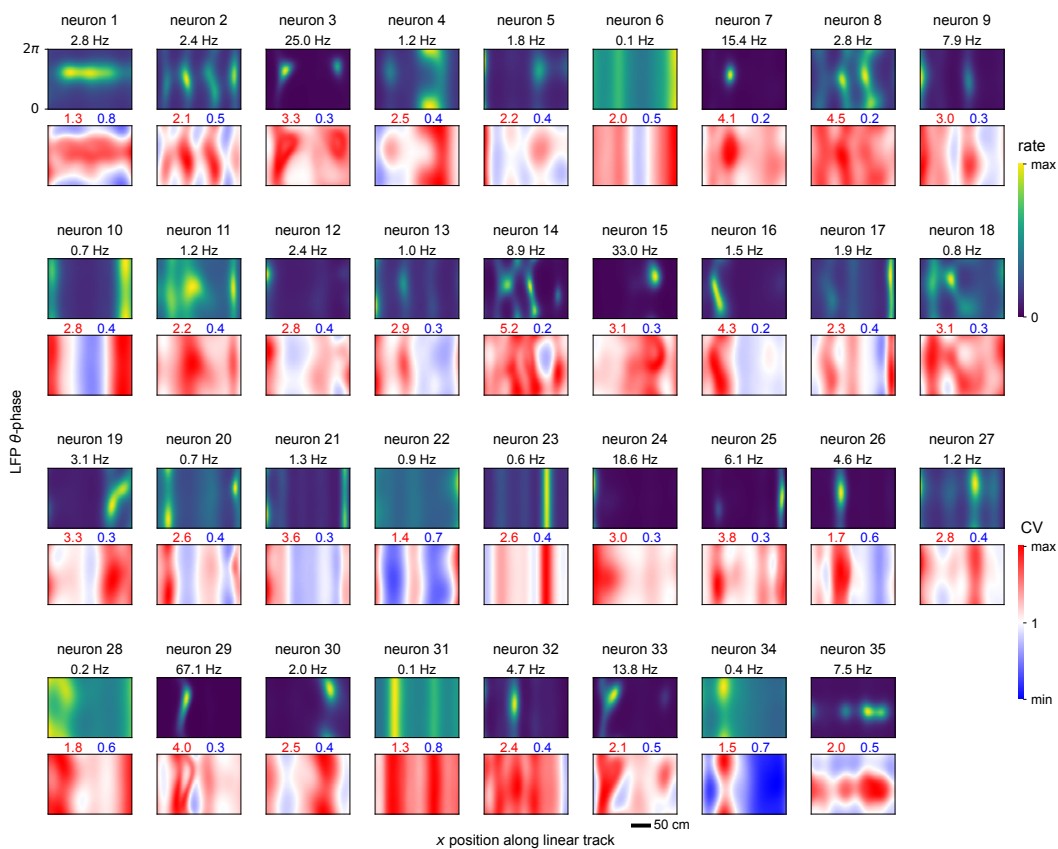

Figure 11: **Tuning curves of place cells for rat running right-to-left.** Heat maps show posterior mean values of spike train statistics computed from the conditional ISI distribution samples.

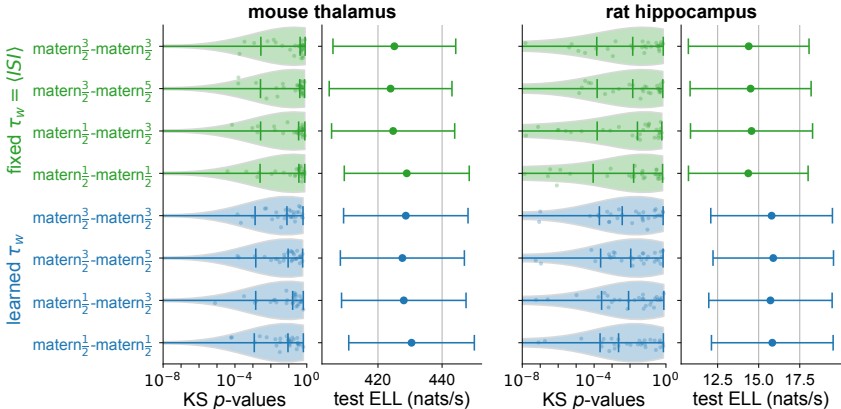

Figure 12: **Comparison of kernel selection for the non-renewal process.** We show measures of model fits to real data similar to the model in the main text for various temporal kernel choices in the format $k(\tilde{\tau})$-$k(\tilde{\boldsymbol{\Delta}})$, and fixed or learned time warping scale $\tau_w$ (green or blue, respectively).