# OpenReview forum: "Bayesian nonparametric (non-)renewal processes for analyzing neural spike train variability"
_NeurIPS.cc/2023/Conference — NeurIPS 2023 poster_

### Official Review · Reviewer_C2R4 · 2023-06-28

**Soundness:** 3 good
**Presentation:** 3 good
**Contribution:** 3 good
**Rating:** 5
**Confidence:** 5

**Summary:**

In this paper, the authors introduce the Bayesian nonparametric non-renewal (NPNR) process to model variability in neural spike trains with covariate dependence. The method generalizes modulated renewal processes using sparse variational Gaussian processes. Tested on synthetic data, as well as mouse head direction cell data and rat hippocampal place cell data, NPNR shows its superiority in terms of capturing interspike interval statistics and predictive power.

**Strengths:**

+ Bayesian nonparametric non-renewal (NPNR) process to model variability in neural spike trains;
+ Validation of the approach on synthetic data and mouse head direction cell data and rat hippocampal place cell data;


**Weaknesses:**

- There are some aspects that while they may be hard to analyze theoretically at least in the experimental part can be investigated through simulations. For example, how is the inference affected by the dimension of x_t, number of unmeasured or unknown neurons  that can act as perturbations, the number of samples and number of units/channels? I.e., how much data (in space and time) is needed for an efficient inference?
- Within the experimental investigation, it is unclear what ELL values can be considered as good predictions. For example, if NPNR is used for k-step ahead forecasts, how many steps can it achieve with at least 50% accuracy?
- Although this is a minor issue, the reader has some difficulties at seeing all the plots well from Figure 2 in particular the ISI probability plots. I assume they fit well gamma or exponential distribution.
- A major issue is to check the literature and in unbiased way provide an accurate description even if the problems considered by prior work have a different or more advanced setup.


**Questions:**

1. The authors mention that the variational inference framework is scalable. In the synthetic and real-world experiments, the number of neurons/units are relatively small (less than 50). I wonder if there's any limitation of the method to scale up to larger ensemble systems with much more neurons?
2. How does the proposed model perform in terms of predictive power when applied to synthetic data?
3. The experiments on mouse head direction cell data and rat hippocampal place cell data shows the predictive performance of NPNR and baseline models by showing the expected log-likelihood. For the two datasets, the range of ELL for the models vary a lot and is hard to interpret. I wonder what value of ELL can be considered as a good prediction? For example, if NPNR is used for k-step ahead forecasts, how many steps can it achieve with at least 50% accuracy?
4. The manuscript states that “Extending point process models with input-dependent variability has not been widely explored…” Multivariate auto-regressive frameworks and multiple covariates based models have been considered in "A Granger causality measure for point process models of ensemble neural spiking activity." PLoS computational biology 7, no. 3 (2011): e1001110. "Data-driven perception of neuron point process with unknown unknowns." In Proceedings of the 10th ACM/IEEE International Conference on Cyber-Physical Systems, pp. 259-269. 2019. "Variance as a signature of neural computations during decision making." Neuron 69, no. 4 (2011): 818-831. In general the prior work needs to be more exhaustively checked and discussed, as of now it is biased and solely based on one group while there are similar and related works from other groups.
5. Within the context of multiple neuronal recordings there is always the issue of interference and the problem that we cannot with certainty measure exactly N number of neurons. The activity of N neurons may be influenced by another P neurons so the question is how we can subtract the effect or perturbations in order to accurately model the N neurons and their covariates, etc. This again has been tackled in the neuroscience literature and the authors should check this related problem of understanding neural computations with unknown influences.
6. In the experiments, 1-D, 2-D and 3-D x_t are considered for the NPNR modeling. I wonder if and how the inference can be affected by the dimension of x_t, number of samples and number of units/channels? I.e., how much data (in space and time) is needed for an efficient inference?


**Limitations:**

Not applicable in my opinion, this is a mathematical modeling paper with applications in neuroscience.

---

> ### Author Rebuttal · Authors · 2023-08-08
>
> We thank the Reviewer for their time and helpful feedback. We provide brief comments on the mentioned weaknesses in the relevant question.
>
> *1. The authors mention that the variational inference framework is scalable...*
>
> The datasets in this work each contain around 30 selected neurons that are fitted simultaneously and have 1 to 2 million time steps in the training set, giving around 30-60 million data points in total (see appendix C.3). Most recent probabilistic models applied to neural data take in 100s of neurons but with binned spike counts, reducing the temporal length to $O(10^5)$ typically (“Scalable Bayesian GPFA with automatic relevance determination and discrete noise models”, Jensen et al. 2021; “Linear Time GPs for Inferring Latent Trajectories from Neural Spike Trains”, Dowling et al. 2023). Hence overall, in terms of number of data points this qualifies as a scalable method by recent standards.
>
> For the camera-ready version, we will add these numbers to Section 4.2 to explicitly demonstrate the scale of our datasets.
>
> *2. How does the proposed model perform in terms of predictive power when applied to synthetic data?*
>
> When extending the synthetic experiment on 5 different held-out synthetic datasets:
>
> | model | test ELL (nats / s) |
> |:--------:|:--------:|
> |  Poisson   | $40.55 \pm 2.09$  |
> |  Gamma renewal   | $47.07 \pm 2.10$ |
> |  conditional Poisson   | $40.82 \pm 2.22$ |
> |  NPNR (ours)    | $44.93 \pm 2.30$ |
>
> Note the Gamma renewal process is within model class for 3 of the synthetic neurons. We will include these numerical results in the camera-ready version in the appendix.
>
> *3. The experiments on mouse head direction cell data and rat hippocampal place cell data...*
>
> The two datasets are very different in terms of the spiking statistics when comparing the figures 3 and 4, hence the differences in the ELL values across the two datasets is consistent. Such numbers nevertheless can be compared in a relative sense within datasets. The test ELLs are also useful for fair comparison of predictive powers for qualitatively different baseline models (Ref. 2, “Construction and analysis of non-Poisson stimulus-response models of neural spiking activity”, Barbieri et al. 2001; Ref. 60, “Time-rescaling methods for the estimation and assessment of non-Poisson neural encoding models”, Pillow 2009; Ref. 17, “Non-parametric generalized linear model”, Dowling et al. 2020).
>
> Our model requires autoregressive sampling of posterior spike trains as seen in figures 3C and 4C, with each draw having a different CIF. The k-step ahead forecast you proposed is therefore not clear to us; prediction accuracy of the held-out spikes after k steps will vary across different posterior spike train samples.
>
> We will point out the difference in ELL values between the two datasets more explicitly in the camera-ready paper with a brief discussion similar to the above.
>
> *4. The manuscript states that...*
>
> We would like to thank the Reviewer for pointing out these works, some of which were not known to us.
>
> - **Kim et al. 2011**: a parametric CIF with a fixed interval history dependence, with a focus on Granger causality structure inference
>
> - **Yang et al. 2019**: a parametric GLM framework with unobserved or latent inputs, while considering the dependence of the spike-history on the external/unobserved covariates
>
> - **Churchland et al. 2011**: great example of the relevance of studying neural variability for neural coding
>
> For the camera-ready paper, we will add these papers to the literature review in the relevant introduction paragraphs with appropriate discussion. Our contribution remains unique in the scalable Bayesian nonparametric modeling of spike-history dependence with emphasis on covariate-dependent modulation of instantaneous spiking variability, the autoregressive dependence in terms of interspike intervals rather than spike-train windows, and finally our perspective on the connection to conditional ISI distributions when conceptualizing spiking variability extracted with our model. We will qualify the incriminated sentence in our Introduction accordingly.
>
> *5. Within the context of multiple neuronal recordings...*
>
> We agree strongly with the Reviewer on the importance of investigating these issues.
>
> The effect of ignoring subsets of neurons generally leads to apparent noise correlations in the neurons we record, which we can capture by augmenting our model with latent variables shared across the population. Overall, capturing neural correlations is orthogonal to our main contribution of modeling the single neuron responses in a more flexible manner, and hence beyond the scope of this paper. Another related tangential topic is the downstream effect of spike sorting contamination (e.g. Ref. 86, “Assessing goodness-of-fit in marked point process models of neural population coding via time and rate rescaling”, Yousefi, A. et al. 2020).
>
> *6. In the experiments, 1-D, 2-D and 3-D x_t are considered for the NPNR modeling...*
>
> In this study, our synthetic data has 2D inputs $x_t$ and has a million time points, and we were able to accurately recover the ground truth (figure 2). Real data had either 1D or 3D $x_t$ and ~2 million time points. Based on the synthetic experiment, we can be quite confident that this was in a sufficient data regime. Our method builds on sparse variational Gaussian processes, and hence inherits the convergence properties and data scaling from such models (Ref. 63, “Efficient, adaptive estimation of two-dimensional firing rate surfaces via Gaussian process methods”, Rad and Paninski 2010; “Rates of Convergence for Sparse Variational Gaussian Process Regression”, Burt et al. 2019; and the references in appendix B.1 on technical papers introducing sparse variational GPs).
>
> We will add and expand the relevant references in appendix B.1 to address these questions for the camera-ready paper, as well as add this discussion topic to Section 3.2 with the relevant references.

---

### Official Review · Reviewer_Mj1g · 2023-07-04

**Soundness:** 3 good
**Presentation:** 4 excellent
**Contribution:** 4 excellent
**Rating:** 8
**Confidence:** 4

**Summary:**

This paper proposes the Bayesian nonparametric non-renewal process (NPNR) for inferencing both neural spiking intensity and variability. The tuning curve is based on a sparse variational Gaussian process (GP) prior, considering both spatial and temporal factors. They compare NPNR with other competitors on a synthetic dataset showing the capability of NPNR in inferencing the rate map and renewal density. On the two real-world neural datasets, they show that NPNR outperforms lots of competitors in terms of event prediction and interspike interval (ISI) recovery by different statistics combined with visualizations.

**Strengths:**

* Clear logical flow and presentation. Key maths are derived elegantly with lots of necessary details in the Appendix.
* Both synthetic and real-world experiments are good and solid, and also supported by the code.
* The literature review by the authors are exhaustive so that the comparison between different kind of models are clear and in detail.
* The idea is new and intuitive, both preserve the interpretability and are not too simple.

**Weaknesses:**

* I'm very excited when seeing the model part. But when I get to Section 3.2, I feel a bit pity that we still need to do time discretization (convert the continuous timestamps TPP data to spike counts in time bins).

**Questions:**

* I think Eq. 15 should be $=$ rather than $\propto$.
* I'm wondering if this model can report a predictive log-likelihood using the mean as the estimation for the intensity in each time bin. In such a case, we can compare this model with other models (especially the simple GLM) to show that the proposed NPNR outperforms GLM? I'm expecting that this model will be slow but if the firing rates (tuning curve) recovery is better than GLM, the predictive log-likelihood (which is actually a golden criterion in neural latent variable models) should be better.
* Can this model get information on the causal relationships between neurons? Is this model only dependent on time $t$ and external input $\boldsymbol x$, but does not consider influences between neurons? From my understanding, this is not a latent variable model doing information extraction from coupled neurons (like dimensionality reduction), but getting the firing rate for each neuron, and the firing rate is mainly affected by the neuron itself and the external input $\boldsymbol x$.

**Limitations:**

/

---

> ### Author Rebuttal · Authors · 2023-08-08
>
> We thank the reviewer for their time and helpful feedback on the work.
>
> **Q&A**
>
> *1. I think Eq. 15 should be = rather than ∝.*
>
> This would indeed be true if the denominator (normalization constant) in equation 20 (appendix A) was 1. For a Poisson process with $t_i = 0$ (i.e. the most recent spike happened at time 0), this is true and the equality holds. In general, this does not hold and we need to compute the normalization constant which leads to valid ISI densities as plotted in the paper.
>
> *2. I'm wondering if this model can report a predictive log-likelihood using the mean as the estimation for the intensity in each time bin. In such a case, we can compare this model with other models (especially the simple GLM) to show that the proposed NPNR outperforms GLM? I'm expecting that this model will be slow but if the firing rates (tuning curve) recovery is better than GLM, the predictive log-likelihood (which is actually a golden criterion in neural latent variable models) should be better.*
>
> In fact, the expected log likelihood (ELL) on the test sets is exactly the predictive log-likelihood you mention (see Section 4.2), defined as
>
> $ \text{ELL} = \sum_n \mathbb{E}_{q(\mathbf{f}_n)} [ \log p(\mathbf{y}_n | \mathbf{f}_n) ] $
>
> Instead of the predictive log-likelihood, we use the term test ELL as we are working in a variational framework rather than with a maximum likelihood approach. This metric also incorporates the variational posterior uncertainty in the intensity estimation in each bin, similar to metrics used in the Gaussian process literature (“Scalable Exact Inference in Multi-Output Gaussian Processes”, Bruinsma et al. 2020; “Dual Parameterization of Sparse Variational Gaussian Processes”, Adam et al. 2021).
>
> The simple GLM (and more advanced variants of it) are actually already presented in figures 2, 3 and 4. These appear under the name “conditional Poisson” or “cond. P” for short (see Section 4.2 first paragraph last sentence). We apologize for the slightly unorthodox nomenclature, but this is necessary as we are comparing many variants and extensions of the GLM/SRM family (see Section 2.1.2). The Poisson GLM model has been extended with different kinds of spike-history filters, hence we also add the type of filter to the name. For example, “RC cond. P” is the radial cosine conditional Poisson in figures 3A and 4A, the classical GLM as referred to in the literature (Ref. 80, “Capturing the dynamical repertoire of single neurons with generalized linear models”, Weber and Pillow 2017).
>
> Since we run all baseline models using Gaussian process mappings in the same variational framework (see appendix B.2, B.3 and B.4), we also compute the test ELL in a similar way to our method (conventional GLMs use a maximum likelihood linear feature model for the input-output mapping). Since this metric is computed over the whole population, by adding latent input dimensions we would arrive at the predictive likelihood that you mentioned for latent variable models (although this is not done in this study due to the focus of this work).
>
> *3. Can this model get information on the causal relationships between neurons? Is this model only dependent on time $t$ and external input $x$, but does not consider influences between neurons? From my understanding, this is not a latent variable model doing information extraction from coupled neurons (like dimensionality reduction), but getting the firing rate for each neuron, and the firing rate is mainly affected by the neuron itself and the external input $x$.*
>
> Generally, causal relationships are very hard or impossible to estimate from statistics alone (see Ref. 42, “Inferring structured connectivity from spike trains under negative-binomial generalized linear models”, Linderman et al. 2015). However, we could capture correlations between neurons through latent variables that would co-modulate the CIFs of individual neurons in the population. Alternatively, we could include the recent spiking history of other neurons in the input covariates in addition to the self-history of a neuron, similar to the GLM framework (“Spatio-temporal correlations and visual signaling in a complete neuronal population”, Pillow et al. 2008; “Non-parametric generalized linear model”, Dowling et al. 2020). In the NPNR framework the latter would naively lead to a significant increase in the input dimensions, and one interesting further work direction is to encode population spike-history in an efficient way.
>
> Indeed, the firing properties of each neuron in this study are only affected by its own history and the external covariates/inputs. For demonstrating our contribution, which is a flexible spike train noise model, this suffices and indeed improves over SOTA methods applied in the same manner.
>
> **Weaknesses and limitations**
>
> *I'm very excited when seeing the model part. But when I get to Section 3.2, I feel a bit pity that we still need to do time discretization (convert the continuous timestamps TPP data to spike counts in time bins).*
>
> We agree with the reviewer and indeed drafts of this work were considering continuous time techniques that are faithful to the continuous-time formulation. However, exact methods for continuous time processes often rely on parametric assumptions that we try to alleviate (“Variational Inference for Gaussian Process Modulated Poisson Processes”, Lloyd et al. 2015; Ref. 18, “Temporal alignment and latent Gaussian process factor inference in population spike trains”, Duncker and Sahani 2018), or use MCMC, generalized thinning and other techniques (Ref. 74, “Gaussian process modulated renewal processes”, Teh and Rao 2011) that are not as straightforward to scale up to millions of time steps as in our work using stochastic variational inference.
>
> Do note that the discretization is done at 1 ms, which for neural data that has been sorted properly (no significant spike contamination) should not contain more than one spike (a binary time series).

---

> > ### Comment · Reviewer_Mj1g · 2023-08-16
> >
> > Thank you very much for your detailed replies. I'm still satisfied with this paper and I think the answers have solved most of my concerns.

---

### Official Review · Reviewer_hah4 · 2023-07-05

**Soundness:** 3 good
**Presentation:** 3 good
**Contribution:** 2 fair
**Rating:** 6
**Confidence:** 3

**Summary:**

The authors proposed a scalable Bayesian approach which generalizes modulated renewal processes using sparse variational Gaussian processes. They applied the proposed method to simulated and two real neural datasets and showed that the proposed method is effective on these datasets and outperforms other baseline methods.

**Strengths:**

The paper is well written. The authors have done extensive experiments to show the effectiveness of the proposed method.

**Weaknesses:**

The proposed method doesn't incorporate any latent variables in the model. (see more in the questions section below.)

**Questions:**

- Method section: all the formulas are described for 1 neuron. It might be clearer to write likelihood and loss function in terms of multiple neurons.

- I wonder if the authors have done any comparisons to the parametric methods? e.g. Gao Y*, Archer E*, Paninski L, Cunningham JP (2016) Linear dynamical neural population models through nonlinear embeddings. NIPS 2016.

- The proposed method doesn't incorporate any latent variables. It might be worth adding the latents to discover useful representations from the data and fit the data variability better. I wonder if the model would fit data worse if miss one covariate in the inputs? (e.g. only includes location and direction in covariate not theta phase in the hippocampus data.) I feel that adding latents might help with this as well.

**Limitations:**

The authors have discussed the limitations and future work of their proposed method.

---

> ### Author Rebuttal · Authors · 2023-08-08
>
> We thank the Reviewer for their time and helpful feedback.
>
> *1. Method Section: all the formulas are described for 1 neuron. It might be clearer to write likelihood and loss function in terms of multiple neurons.*
>
> The current work inherently models each neuron separately while fitting the neural population simultaneously, hence the likelihood for all neurons is simply the sum of the log likelihoods in equation 18
>
> $\mathcal{L} = \sum_n \mathbb{E}\_{ q( f_n \| u_n) } \left[ - y_{nt} f_{nt} + \Delta t \sum_{t=1}^T e^{f_{nt}} \right] + D_{\text{KL}}( q(\mathbf{u}_n) |\ p(\mathbf{u}_n) ) )$
>
> where in $f_{nt}$ etc. we add the neuron index to the quantities. We will modify this in the camera-ready version to make this more explicit. As you suggest below, introducing latent variables would provide the ability to capture neural (noise) correlations. See our response to question 3 below for a brief discussion on this topic.
>
> *2. I wonder if the authors have done any comparisons to the parametric methods? e.g. Gao Y\*, Archer E\*, Paninski L, Cunningham JP (2016) Linear dynamical neural population models through nonlinear embeddings. NIPS 2016.*
>
> (Gao et al. 2016) is a model for spike counts with parametric noise models (Poisson and generalized count distributions). Our work is in the realm of spike trains where we have essentially binary time series after time discretization (see Section 3.2). We discussed related works in both realms, see the introductory paragraphs starting at line 40 to line 61, i.e. last paragraphs before the “contribution” paragraph. In this case of working with point processes/spike trains, spike count models are not appropriate as the only “spike count distribution” in a time bin is a Bernoulli distribution. Note that the main contribution of this paper is the flexible statistical modeling of the spiking process of a neural, or in machine learning terms flexible modeling of the noise distribution, per neuron. Works like (Gao et al. 2016) are latent variable models that model neural correlations despite having the same separable structure across neurons in the noise model.
>
> In terms of parametric models, we conducted an extensive comparison of current parametric models as GLMs/SRMs under the name conditional processes (Ref 80, “Capturing the dynamical repertoire of single neurons with generalized linear models”, Weber and Pillow 2017; Ref. 17, “Non-parametric generalized linear model”, Dowling et al. 2020), renewal processes and other variants described in Section 2 (details in appendix B.2-3 and C). Results are shown for various models in figures 3 and 4, with the abbreviations described in the last sentence of the first paragraph in Section 4.2 (which include the simple GLMs as “RC cond. P” or radial cosine conditional Poisson processes).
>
> In the camera-ready version, we will include (Gao et al. 2016) in the introductory paragraphs starting at line 40 to line 61 and more explicitly mention the lack of neural correlations in Section 3.1 (similar to the discussion in section C.1, “Vectorization over neurons”).
>
> *3. The proposed method doesn't incorporate any latent variables. It might be worth adding the latents to discover useful representations from the data and fit the data variability better. I wonder if the model would fit data worse if miss one covariate in the inputs? (e.g. only includes location and direction in covariate not theta phase in the hippocampus data.) I feel that adding latents might help with this as well.*
>
> Indeed, we agree that adding latent variables and thus making the model suitable for analyzing neural noise correlations is a major avenue for further research (see Section 5.1). Our current contribution is a model that is able to capture single neuron statistics with a sufficiently flexible noise model, which shows up as explaining all the data variance as measured by usual distribution comparison methods for point processes (see the KS plots figures 2C, 3B and 4B). However, such measures do not take into account neural correlations, and extensions to the KS framework have been proposed as in Ref. 28 (“Applying the multivariate time-rescaling theorem to neural population models”, Haslinger and Pipa 2011). Importantly, our current implementation is suitable for latent variable augmentations as we fit neurons in parallel like a coherent neural population (see appendix C.1 and the JAX implementation provided).
>
> Adding more covariates generally will improve the test ELL/predictive performance from the following perspective: our model can capture arbitrary spike train noise models (in theory), and the signal that is modeled is the modulation of the flexible noise distribution by those covariates. Adding more covariates allows the model to discover more fluctuating signals since we have more freedom if the covariates are uncorrelated. When the signal can account for more of the data variance, the noise model inferred can be sharper, i.e. more precise, and hence this will mathematically lead to higher log likelihoods. The extreme case here would be a noise model leading to conditional ISI densities that are delta distributions at the average ISI given some covariates (such models are approximated by Gamma renewal processes with shape parameters well above 1).
>
> In the Reviewer’s example, including theta allows one to capture oscillations in the firing output of neurons with the theta rhythm (one can see the wiggled curve in figure 4C, note the contrast to smoother rates in figure 3C where no fast oscillating covariates as theta phase are included). If we fit the same model leaving out the theta phase covariate, we obtain similar goodness-of-fit measures (as the flexible noise model/ISI densities adapt to account for the reduction in signal) but lower test ELLs:
>
> | model | test ELL (nats / s) |
> |:--------:|:--------:|
> | $x$ position + head direction  | $13.92 \pm 3.90$  |
> | $x$ position + head direction + theta phase | $14.39 \pm 4.11$ |

---

> > ### Comment · Reviewer_hah4 · 2023-08-18
> >
> > Thank the authors for their detailed response! They have addressed my concerns, so I increase the rating to 6.

---

### Official Review · Reviewer_yMLW · 2023-07-06

**Soundness:** 3 good
**Presentation:** 3 good
**Contribution:** 3 good
**Rating:** 7
**Confidence:** 3

**Summary:**

The variability of neural data is widely observed in many neuroscience experiments. Using statistical model to capture the variability structure plays an essential role in understanding neural computations. Generally, the variability of neural data is a result of non-stationary activities and dependencies on behavioral covariates. To tackle these challenges, the authors proposes a scalable Bayesian approach generalizing modulated renewal processes (NPNR) to analyze neural data variability. They develops a nonparametric generalization of modulated renewal processes beyond renewal order, which makes their method flexibly model irreducible “intrinsic” neural stochasticity and input-dependent variability. Furthermore, the authors apply stochastic variational inference and can fit the model to long time neural data given cubic time complexity in the number of inducing points. The performace of NPNR is evaluated on both synthetic and real neural datasets.

**Strengths:**

* The proposed method can model two types of neural variability: (1) capturing spiking statistics from non-stationary data; (2) capturing modulation by behavioral covariates.
* To achieve the desired non-stationarity, the proposed method uses time warping on $\tau$, which avoids the use of non-stationary kernls and maintains the ability to draw samples by pathwise conditioning.
* The proposed inference method provides an elegant approach to determine the spike-history dependence in ISI statistics.
* The authors' exposition of their motivation, contribution, and conclusions from the experiments are comprehensive and clear.
* The proposed method would provide an important set of contributions to the field of neural coding.

**Weaknesses:**

The proposed method is clearly written and well supported by experiments. I have nothing further to add here.

**Questions:**

* The proposed method captures ISI statistics using a spatio-temporal GP prior over CIF. Could the Neural Temporal Point Process (NTPP) perform similarly to your method in capturing ISI statistics? The CIF in NTPP is usually modeled by neural networks, and could this be more powerful to represent ISI distributions and capture ISI statistics?

---

> ### Author Rebuttal · Authors · 2023-08-08
>
> We thank the Reviewer for their time and helpful feedback.
>
> *1. The proposed method captures ISI statistics using a spatio-temporal GP prior over CIF. Could the Neural Temporal Point Process (NTPP) perform similarly to your method in capturing ISI statistics? The CIF in NTPP is usually modeled by neural networks, and could this be more powerful to represent ISI distributions and capture ISI statistics?*
>
> Indeed, we are also familiar with the NTPP works and have considered them as candidates. However, the Gaussian process approach offers a few unique properties that are difficult to achieve with neural networks:
> - Automatic relevance determination of the lagged ISI dimensions by learning temporal kernel lengthscales (see figures 2B, 3D, 4D and 6A)
> - Principled regularization using the evidence lower bound and smoothness constraints on the CIF dependence on covariates $\mathbf{x}_t$ and past ISIs $\mathbf{\Delta}_t$ (e.g. figure 12 comparing against different kernels that impose different levels of differentiability of the CIF (i.e. choosing different order Matern kernels)
>
> In general, any method that involves flexible modeling of the CIF directly is closely related to the main idea/contribution of this work. Our work introduces nonparametric autoregressive dependence on past ISIs, which adds $K$ more input dimension compared to baselines which handle past dependencies using parametric spike history filters or renewal assumptions. For this reason, kernel methods such as Gaussian processes provide an elegant framework for handling this without overfitting as much as neural networks.
>
> A detailed comparison study of neural networks replacing the Gaussian process for modeling the CIF is out of scope for the current work and is more related to the general argument of Gaussian processes versus deep neural networks, which is an entire subfield on its own. Small examples of neural networks overfitting in cases when Gaussian processes do not are for instance given in Ref. 43 (“A universal probabilistic spike count model reveals ongoing modulation of neural variability”, Liu and Lengyel 2021). Additionally, there are many influential works investigating the relationship between these two classes (“Approximate Inference Turns Deep Networks into Gaussian Processes”, Khan et al. 2019 ; “Wide Feedforward or Recurrent Neural Networks of Any Architecture are Gaussian Processes”, Yang 2019).
>
> An interesting case mentioned in our work is Ref.54 (“Fully Neural Network based Model for General Temporal Point Processes”, Omi et al. 2019), where the authors chose to model the cumulative hazard function (integral of the CIF) for temporal point processes. This requires a monotonic function approximator, which is most easily done with constrained neural networks and difficult to achieve with Gaussian process priors.
>
> In the camera-ready paper, we will expand on the comparison to neural network approaches similar to the discussion above, and add relevant references on these topics.

---

> > ### Comment · Reviewer_yMLW · 2023-08-19
> >
> > I thank the authors to answer my question. It's a good work to discuss neural data variability.

---

### Official Review · Reviewer_3xxY · 2023-07-06

**Soundness:** 3 good
**Presentation:** 3 good
**Contribution:** 3 good
**Rating:** 6
**Confidence:** 2

**Summary:**

This paper proposes a Bayesian nonparametric approach using modulated renewal process to model neural spike train, and capable of modeling the covariability. The method includes a nonparametric priors on conditional interspike interval distribution and automatic relevance determination for lagging interspike interval based on renewal order. The method is evaluated on one synthetic data and two real datasets on animal navigation. It demonstrates better performance than the current SOTA baselines in its capability of capturing the interspike interval statistics.

**Strengths:**

1. Motivation: the paper is well-motivated, modeling the spike train statistics is an important question, and the interspike interval statistics is an important property of neural dynamics, and potentially leads to identification of cell types, and functionality, etc. Designing a model that captures this property well is important.

2. Method: it uses Bayesian nonparameteric approach that could fit complex data structures and patterns well, and it could infer the spike-history dependence using a data-driven approach.

3. Results: the method demonstrate better accuracy than the current SOTA methods in multiple tasks and datasets.

**Weaknesses:**

1. Method: the model requires hyperparameter tuning of critical components includes $\tau_w$  $K$, which might be hard to optimize, given its variability across neurons and datasets.

2. Evaluation: the scalability of the method is a major concerns, as the datasets evaluated in this paper only has 9 neurons, ~30 units in each datasets. It's important to show how well the model performs in larger neural datasets.

3. Complexity and computational cost: add evaluations based on the cost and speed of the proposed model and other baselines.



**Questions:**

1. Give a detailed introduction about the parameters that would be optimized under eqn 18. and list other hyper-parameters for reproducibility.

2. Evaluate the method on larger scale neural datasets.

**Limitations:**

1. There is no potential negative societal impact of their work.
2. The limitations about scalability of the proposed approach should be carefully addressed.

---

> ### Author Rebuttal · Authors · 2023-08-08
>
> We thank the Reviewer for their time and helpful feedback.
>
> **Q&A**
>
> *1. Give a detailed introduction about the parameters that would be optimized under eqn 18. and list other hyper-parameters for reproducibility.*
>
> The NPNR model parameters consist of:
> - Gaussian process hyperparameters (kernel hyperparameters $\theta_{GP}$, variational posterior mean $\mu$ and covariance $\Sigma$, inducing point locations $Z$, Gaussian process mean function parameters $\tau_m$ and $b_m$). All these parameters are learned with gradient descent
> - Time warping timescale $\tau_w$. This parameter is fixed in our experiments to the (empirical) mean ISI, but we also compared the cases where we jointly learn this parameter with the rest (see appendix figure 12)
> - Maximum lagging ISI order $K$ (integer). This hyperparameter is chosen in advance to a number we expect to be sufficiently large (in general, it is unlikely to have many neurons with significant history dependence beyond the past 2 ISIs based on studies of single neuron modeling). Indeed, figure 3D and 4D show this applies to most neurons in both datasets
>
> For the camera-ready version, we will provide a short description at the end of the inference Section summarizing all the parameters and hyperparameters, stating which are learned and which are not.
>
> *2. Evaluate the method on larger scale neural datasets.*
>
> The datasets in this work each contain around 30 selected neurons that are fitted simultaneously and have 1 to 2 million time steps in the training set (1 ms resolution), giving around 30-60 million data points in total (see appendix C.3). Most recent probabilistic models applied to neural data take in 100s of neurons but often with binned spike counts, reducing the temporal length to O(10^5) to O(10^6) in typical studies (e.g. “Scalable Bayesian GPFA with automatic relevance determination and discrete noise models”, Jensen et al. 2021; “Linear Time GPs for Inferring Latent Trajectories from Neural Spike Trains”, Dowling et al. 2023). Hence overall, in terms of number of data points this qualifies as a scalable method by recent standards.
>
> We fit did not fully utilize the 11 GB GPU RAM that we ran the experiment on. With the same temporal batch size and other settings, we generally observe OOM issues around ~80 neurons, which can be increased even further by reducing the temporal batch size. We encourage the reviewer to try out the JAX implementation that is provided, which also provides fully reproducible runs of the experiments presented in this study.
>
> For the camera-ready version, we will add these numbers (50-60 millions data points and total time points from appendix C.3) to Section 4.2 to explicitly demonstrate the scale of our datasets, which can be misleading when just looking at the number of neurons (~30 neurons).
>
> **Weaknesses and limitations**
>
> *1. Method: the model requires hyperparameter tuning of critical components includes tau_w, K, which might be hard to optimize, given its variability across neurons and datasets.*
>
> The time warping process is chosen based on inductive biases from the neuroscience literature (see Section 3.1). We fix $\tau_w$ instead of learning it as this leads to automatic relevance determination in the temporal kernel timescales (see figure 6A). In appendix figure 12, we show that the loss of performance due to fixing this parameter is minimal.
>
> Choosing $K$ is in a sense similar to choosing the number of layers in a deep neural network, or the dimensionality of latent spaces in latent variable models like VAEs. A rigorous approach would involve a grid search, but in the spirit of Bayesian models one can use a single high capacity model and perform automatic relevance determination (“Scalable Bayesian GPFA with automatic relevance determination and discrete noise models”, Jensen et al. 2021; “Bayesian Gaussian Process Latent Variable Model”, Titsias and Lawrence 2010) to get rid of redundant degrees of freedom as shown in figure 2B, 3D and 4D.
>
> In the camera-ready paper, we will elaborate on the issue of hyperparameter selection and tuning as discussed above in Section 3.2, and include relevant references to similar procedures done in the literature. We will explicitly mention and discuss all our hyperparameter tuning results (appendix figure 12 and automatic relevance determination figure 2B, 3D and 4D) in the light of hyperparameter tuning.
>
> *2. Evaluation: the scalability of the method is a major concerns, as the datasets evaluated in this paper only has 9 neurons, ~30 units in each datasets. It's important to show how well the model performs in larger neural datasets.*
>
> See response to question 2 in Q&A.
>
> *3. Complexity and computational cost: add evaluations based on the cost and speed of the proposed model and other baselines.*
>
> The stochastic sparse variational Gaussian processes (Hensman et al. 2013) has computational complexity $O(N T M^2 + N M^3)$, where $M$ is the number of inducing points, $N$ the number of output dimensions (neurons) and $T$ is the temporal batch size. As we can shorten the batch size $T$, the predominant factor is $O(M^3)$ as we mention in Section 3.2 in the main text. Our method has the same computational scaling as stochastic sparse variational Gaussian processes (“Gaussian processes for big data”, Hensman et al. 2013), which can be applied to very large datasets due to mini-batching. Baseline models from the literature are adapted to use the same underlying SVGP machinery to model the input-output mapping or spike-history dependence, and hence inherits the similar complexity scaling up to multiplicative factors. A detailed discussion of model implementation and inference is presented in appendix B.2-5, where we present a large amount of details not suitable for the main text.
>
> For the camera-ready paper, we will extend the discussion on computational complexity in Section 3.2 and appendix B.2-5, with the full $O(N T M^2 + N M^3)$ expression.

---

> > ### Comment · Reviewer_3xxY · 2023-08-17
> >
> > Thanks the authors for providing more details for their methods, especially related to the scalability and hyperparameter tuning. I improved my score accordingly.

---

### Decision · Program_Chairs · 2023-09-21

**Decision:**

Accept (poster)

**Comment:**

This is a solid piece of work using time-warped Gaussian processes to model the conditional intensity function / inter-spike interval distribution. While the approach draws on standard machinery, it shows promising performance on a variety of multi-neuronal spike train recordings. The experimental validation is thorough and the presentation is clear. This paper is a valuable contribution to the statistical neuroscience literature.